# What do Vision Transformers Learn? A Visual Exploration

## Abstract

Vision transformers (ViTs) are quickly becoming the de-facto architecture for computer vision, yet we understand very little about why they work and what they learn. While existing studies visually analyze the mechanisms of convolutional neural networks, an analogous exploration of ViTs remains challenging. In this paper, we first address the obstacles to performing visualizations on ViTs. Assisted by these solutions, we observe that neurons in ViTs trained with language model supervision (e.g., CLIP) are activated by semantic concepts rather than visual features. We also explore the underlying differences between ViTs and CNNs, and we find that transformers detect image background features, just like their convolutional counterparts, but their predictions depend far less on high-frequency information. On the other hand, both architecture types behave similarly in the way features progress from abstract patterns in early layers to concrete objects in late layers. In addition, we show that ViTs maintain spatial information in all layers except the final layer. In contrast to previous works, we show that the last layer most likely discards the spatial information and behaves as a learned global pooling operation. Finally, we conduct large-scale visualizations on a wide range of ViT variants, including DeiT, CoaT, ConViT, PiT, Swin, and Twin, to validate the effectiveness of our method.

| Edges | Textures | Patterns | Parts | Objects |
|---|---|---|---|---|

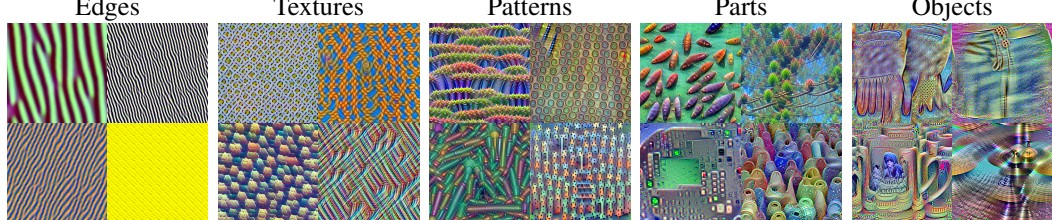

Figure 1: **The progression for visualized features of ViT B-32.** Features from early layers capture general edges and textures. Moving into deeper layers, features evolve to capture more specialized image components and finally concrete objects.

## 1 Introduction

Recent years have seen the rapid proliferation of vision transformers (ViTs) across a diverse range of tasks from image classification to semantic segmentation to object detection (Dosovitskiy et al., 2020; He et al., 2021; Dong et al., 2021; Liu et al., 2021; Zhai et al., 2021; Dai et al., 2021). Despite their enthusiastic adoption and the constant introduction of architectural innovations, little is known about the inductive biases or features they tend to learn. While feature visualizations and image reconstructions have provided a looking glass into the workings of CNNs (Olah et al., 2017; Zeiler & Fergus, 2014; Dosovitskiy & Brox, 2016), these methods have shown less success for understanding ViT representations, which are difficult to visualize. In this work we show that, if properly applied to the correct representations, feature visualizations can indeed succeed on ViTs. This insight allows us to visually explore ViTs and the information they glean from images.

In order to investigate the behaviors of vision transformers, we first establish a visualization framework that incorporates improved techniques for synthesizing images that maximally activate neurons. By dissecting and visualizing the internal representations in the transformer architecture, we find that patch tokens preserve spatial information throughout all layers except the last attention block. The last layer of ViTs learns a token-mixing operation akin to average pooling, such that the classification head exhibits comparable accuracy when ingesting a random token instead of the CLS token.

After probing the role of spatial information, we delve into the behavioral differences between ViTs and CNNs. When performing activation maximizing visualizations, we notice that ViTs consistently generate higher quality image backgrounds than CNNs. Thus, we try masking out image foregrounds during inference, and find that ViTs consistently outperform CNNs when exposed only to image backgrounds. These findings bolster the observation that transformer models extract information from many sources in an image to exhibit superior performance on out-of-distribution generalization (Paul & Chen, 2021) as well as adversarial robustness (Shao et al., 2021). Additionally, convolutional neural networks are known to rely heavily on high-frequency texture information in images (Geirhos et al., 2018). In contrast, we find that ViTs perform well even when high-frequency content is removed from their inputs.

While vision-only models contain simple features corresponding to distinct physical objects and shapes, we find that language supervision in CLIP (Radford et al., 2021) results in neurons that respond to complex abstract concepts. This includes neurons that respond to visual characteristics relating to parts of speech (e.g. epithets, adjectives, and prepositions), a "music" neuron that responds to a wide range of visual scenes, and even a "death neuron" that responds to the abstract concept of morbidity.

Our contributions are summarized as follows:

**I.** We observe that uninterpretable and adversarial behavior occurs when applying standard methods of feature visualization to the relatively low-dimensional components of transformer-based models, such as keys, queries, or values. However, applying these tools to the relatively high-dimensional features of the position-wise feedforward layer results in successful and informative visualizations. We conduct large-scale visualizations on a wide range of transformer-based vision models, including ViTs, DeiT, CoaT, ConViT, PiT, Swin, and Twin, to validate the effectiveness of our method.

**II.** We show that patch-wise image activation patterns for ViT features essentially behave like saliency maps, highlighting the regions of the image a given feature attends to. This behavior persists even for relatively deep layers, showing the model preserves the positional relationship between patches instead of using them as global information stores.

**III.** We compare the behavior of ViTs and CNNs, finding that ViTs make better use of background information and rely less on high-frequency, textural attributes. Both types of networks build progressively more complex representations in deeper layers and eventually contain features responsible for detecting distinct objects.

**IV.** We investigate the effect of natural language supervision with CLIP on the types of features extracted by ViTs. We find CLIP-trained models include various features clearly catered to detecting components of images corresponding to caption text, such as prepositions, adjectives, and conceptual categories.

## 2 RELATED WORK

### 2.1 OPTIMIZATION-BASED VISUALIZATION

One approach to understanding what models learn during training is using gradient descent to produce an image which conveys information about the inner workings of the model. This has proven to be a fruitful line of work in the case of understanding CNNs specifically. The basic strategy underlying this approach is to optimize over input space to find an image which maximizes a particular attribute of the model. For example, Erhan et al. (2009) use this approach to visualize images which maximally activate specific neurons in early layers of a network, and Olah et al. (2017) extend this to neurons, channels, and layers throughout a network. Simonyan et al. (2014); Yin et al. (2020) produce images

which maximize the score a model assigns to a particular class. Mahendran & Vedaldi (2015) apply a similar method to invert the feature representations of particular image examples.

Recent work Ghiasi et al. (2021) has studied techniques for extending optimization-based class visualization to ViTs. We incorporate and adapt some of these proposed techniques into our scheme for feature visualization.

## 2.2 OTHER VISUALIZATION APPROACHES

Aside from optimization-based methods, many other ways to visualize CNNs have been proposed. Dosovitskiy & Brox (2016) train an auxiliary model to invert the feature representations of a CNN. Zeiler & Fergus (2014) use 'deconvnets' to visualize patches which strongly activate features in various layers. Simonyan et al. (2014) introduce saliency maps, which use gradient information to identify what parts of an image are important to the model's classification output. Zimmermann et al. (2021) demonstrate that natural image samples which maximally activate a feature in a CNN may be more informative than generated images which optimize that feature. We draw on some aspects of these approaches and find that they are useful for visualizing ViTs as well.

## 2.3 UNDERSTANDING ViTs

Given their rapid proliferation, there is naturally great interest in how ViTs work and how they may differ from CNNs. Although direct visualization of their features has not previously been explored, there has been recent progress in analyzing the behavior of ViTs. Paul & Chen (2021); Naseer et al. (2021); Shao et al. (2021) demonstrate that ViTs are inherently robust to many kinds of adversarial perturbations and corruptions. Raghu et al. (2021) compare how the internal representation structure and use of spatial information differs between ViTs and CNNs. Chefer et al. (2021) produce 'image relevance maps' (which resemble saliency maps) to promote interpretability of ViTs.

## 3 ViT FEATURE VISUALIZATION

Like many visualization techniques, we take gradient steps to maximize feature activations starting from random noise (Olah et al., 2017). To improve the quality of our images, we penalize total variation (Mahendran & Vedaldi, 2015), and also employ the Jitter augmentation (Yin et al., 2020), the ColorShift augmentation, and augmentation ensembling (Ghiasi et al., 2021). Finally, we find that Gaussian smoothing facilitates better visualization in our experiments as is common in feature visualization (Smilkov et al., 2017; Cohen et al., 2019).

Each of the above techniques can be formalized as follows. A ViT represents each patch $p$ (of an input $x$) at layer $l$ by an array $A_{l,p}$ with $d$ entries. We define a feature vector $f$ to be a stack composed of one entry from each of these arrays. Let $f_{l,i}$ be formed by concatenating the $i$th entry in $A_{l,p}$ for all patches $p$. This vector $f$ will have dimension equal to the number of patches. The optimization objective starts by maximizing the sum of the entries of $f$ over inputs $x$. The main loss is then

$$\mathcal{L}_{\text{main}}(x, l, i) = \sum_p (f_{l,i})_p. \tag{1}$$

We employ total variation regularization by adding the term $\lambda TV(x)$ to the objective. $TV$ represents the total variation, and $\lambda$ is the hyperparameter controlling the strength of its regularization effect. We can ensemble augmentations of the input to further improve results. Let $\mathcal{A}$ define a distribution of augmentations to be applied to the input image $x$, and let $a$ be a sample from $\mathcal{A}$. To create a minibatch of inputs from a single image, we sample several augmentations $\{a_k\}$ from $\mathcal{A}$. Finally, the optimization problem is:

$$x^* = \arg\max_x \sum_k \mathcal{L}_{\text{main}}(a_k(x), l, i) + \lambda TV(a_k(x)). \tag{2}$$

We achieve the best visualizations when $\mathcal{A}$ is $GS(CS(Jitter(x)))$, where $GS$ denotes Gaussian smoothing and $CS$ denotes ColorShift, whose formulas are:

$$GS(x) = x + \epsilon; \;\; \epsilon \sim \mathcal{N}(0, 1)$$

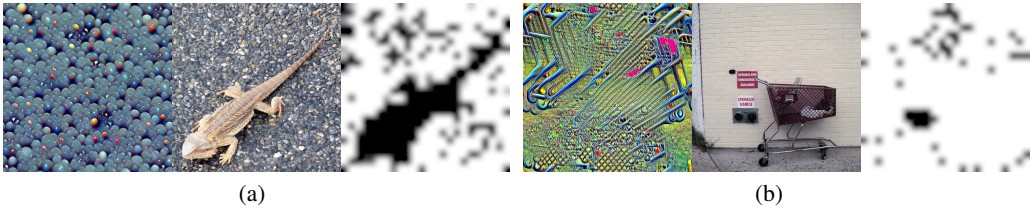

(a)                                                                    (b)

Figure 2: *(a)*: **Example feature visualization from ViT feed forward layer.** *Left:* Image optimized to maximally activate a feature from layer 5. *Center:* Corresponding maximally activating ImageNet example. *Right:* The image's patch-wise activation map. *(b)*: **A feature from the last layer most activated by shopping carts.**

$$CS(x) = \sigma x + \mu; \ \mu \sim \mathcal{U}(-1, 1); \ \sigma \sim e^{\mathcal{U}(-1,1)}.$$

Note that even though $\epsilon$ and $\mu$ are both additive noise, they act on the input differently since $\mu$ is applied per channel (i.e. has dimension three), and $\epsilon$ is applied per pixel. For more details on hyperparameters, refer to Appendix C.

To better understand the content of a visualized feature, we pair every visualization with images from the ImageNet validation/train set that most strongly activate the relevant feature. Moreover, we plot the feature's activation pattern by passing the most activating images through the network and showing the resulting pattern of feature activations. Figure 2(a) is an example of such a visualization. From the leftmost panel, we hypothesize that this feature corresponds to gravel. The most activating image from the validation set (middle) contains a lizard on a pebbly gravel road. Interestingly, the gravel background lights up in the activation pattern (right), while the lizard does not. The activation pattern in this example behaves like a saliency map (Simonyan et al., 2014), and we explore this phenomenon across different layers of the network further in Section 4.

The model we adopt for the majority of our demonstrations throughout the paper is ViT-B16, implemented based on the work of Dosovitskiy et al. (2020). In addition, in the Appendix, we conduct large-scale visualizations on a wide range of ViT variants, including DeiT Touvron et al. (2021a), CoaT Xu et al. (2021), ConViT d'Ascoli et al. (2021), PiT Heo et al. (2021), Swin Liu et al. (2021), and Twin Chu et al. (2021), 38 models in total, to validate the effectiveness of our method. ViT-B16 is composed of 12 blocks, each consisting of multi-headed attention layers, followed by a projection layer for mixing attention heads, and finally followed by a position-wise-feed-forward layer. For brevity, we henceforth refer to the position-wise-feed-forward layer simply as the feed-forward layer. In this model, every patch is always

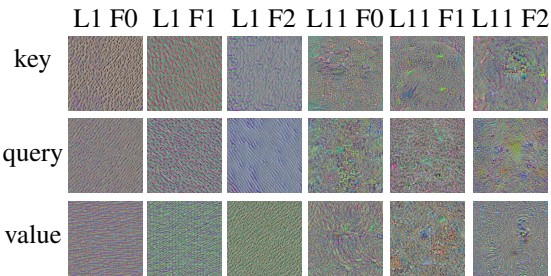

Figure 3: *Left*: **Visualization of key, query, and value.** The visualization both fails to extract interpretable features and to distinguish between early and deep layers. High-frequency patterns and adversarial behavior dominate. *Right*: **ViT feed forward layer**. The first linear layer increases the dimension of the feature space, and the second one brings it back to its initial dimension.

represented by a vector of size 768 except in the feed-forward layer which has a size of 3072 (4 times larger than other layers).

We first attempt to visualize features of the multi-headed attention layer, including visualization of the keys, queries, and values, by performing activation maximization. We find that the visualized feed-forward features are significantly more interpretable than other layers. We attribute this difficulty of visualizing other layers to the property that ViTs pack a tremendous amount of information into only 768 features, (e.g. in keys, queries, and values) which then behave similar to multi-modal neurons, as discussed by Goh et al. (2021), due to many semantic concepts being encoded in a low dimensional space. Furthermore, we find that this behaviour is more extreme in deeper layers. See Figure 3 for examples of visualizations of keys, queries and values in both early and deep layers of

L1 F31   L4 F9   L6 F2   L7 F37   L8 F19   L11 F16

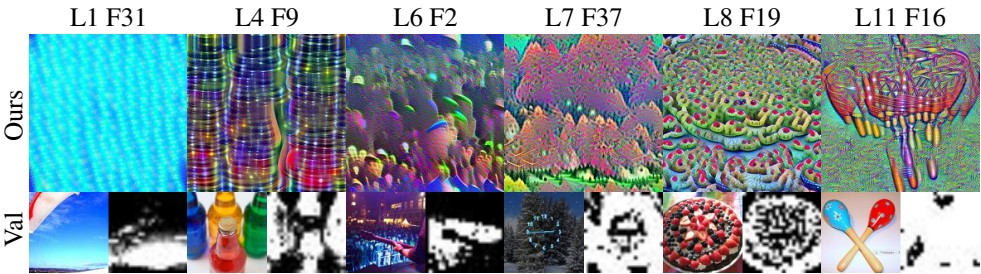

Figure 5: **Feature activation maps in internal layers can effectively segment the contents of an image with respect to a semantic concept.** For each image triple, the visualization on top shows the result of our method, the image on the bottom left is the most activating image from the validation set and the image on the bottom right shows the activation pattern.

the ViT. Inspired by these observations, we visualize the features within the feed-forward layer across all 12 blocks of the ViT. We refer to these blocks interchangeably as layers.

The feed-forward layer depicted in Figure 4 takes an input of size $d = 768$, projects it into a $t = 4$ times higher dimensional space, applies the non-linearity GELU, and then projects back to $d$ dimensional space. Unless otherwise stated, we always visualize the output of the GELU layers in our experiments. We hypothesize that the network exploits these high-dimensional spaces to store relatively disentangled representations. On the other hand, compressing the features into a lower dimensional space may result in the jumbling of features, yielding uninterpretable visualizations.

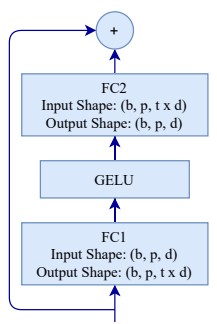

## 4 LAST-LAYER TOKEN MIXING

In this section, we investigate the preservation of patch-wise spatial information observed in the visualizations of patch-wise feature activation levels which, as noted before, bear some similarity to saliency maps. Figure 2(a) demonstrates this phenomenon in layer 5, where the visualized feature is strongly activated for almost all rocky patches but not for patches that include the lizard. Additional examples can be seen in Figure 5 and the Appendix, where the activation maps approximately segment the image with respect to some relevant aspect of the image. We find it surprising that even though every patch *can* influence the representation of every other patch, these representations remain local, even for individual channels in deep layers in the network. While a similar finding for CNNs, whose neurons may have a limited receptive field, would be unsurprising, even neurons in the first layer of a ViT have a complete receptive field. In other words, ViTs *learn* to preserve spatial

Figure 4: **ViT feed forward layer**. The first linear layer increases the dimension of the feature space, and the second one brings it back to its initial dimension.

information, despite lacking the inductive bias of CNNs. Spatial information in patches of deep layers has been explored in Raghu et al. (2021) through the CKA similarity measure, and we further show that spatial information is in fact present in individual channels.

The last layer of the network, however, departs from this behavior and instead appears to serve a role similar to average pooling. We include quantitative justification to support this claim in Appendix section G. Figure 2(b) shows one example of our visualizations for a feature from the last layer that is activated by shopping carts. The activation pattern is fairly uniform across the image. For classification purposes, ViTs use a fully connected layer applied only on the class token (the *CLS* token). It is possible that the network globalizes information in the last layer to ensure that the CLS token has access to the entire image, but because the CLS token is treated the same as every other patch by the transformer, this seems to be achieved by globalizing across *all* tokens.

Based on the preservation of spatial information in patches, we hypothesize that the CLS token plays a relatively minor role throughout the network and is not used for globalization until the last layer. To demonstrate this, we perform inference on images without using the CLS token in layers 1-11,

Table 1: After the last layer, every patch contains the same information. "Isolating CLS" denotes the experiment where attention is only performed between patches before the final attention block, while "Patch Average" and "Patch Maximum" refer to the experiment in which the classification head is placed on top of individual patches without fine-tuning. Experiments conducted on ViT-B16.

| Accuracy | Natural Accuracy | Isolating CLS | Patch Average | Patch Maximum |
|---|---|---|---|---|
| Top 1 | 84.20 | 78.61 | 75.75 | 80.16 |
| Top 5 | 97.16 | 94.18 | 90.99 | 95.65 |

meaning that in these layers, each patch only attends to other patches and not to the CLS token. At layer 12, we then insert a value for the CLS token so that other patches can attend to it and vice versa. This value is obtained by running a forward pass using only the CLS token and no image patches; this value is constant across all input images.

The resulting hacked network that only has CLS access in the last layer can still successfully classify $78.61\%$ of the ImageNet validation set as shown in Table 1. From this result, we conclude that the CLS token captures global information mostly at the last layer, rather than building a global representation throughout the network.

We perform a second experiment to show this last-layer globalization behaviour is not exclusive to the CLS token, but actually occurs across every patch in the last layer. We take the fully connected layer trained to classify images on top of the CLS token, and *without any fine-tuning or adaptation*, we apply it to each patch, one at a time. This setup still successfully classifies $75.75\%$ of the validation set, on average across individual patches, and the patch with the maximum performance achieves $80.16\%$ accuracy (see Table 1), further confirming that the last layer performs a token mixing operation so that all tokens contain roughly identical information. Figure 6 contains a heat-map depicting the performance of this setup across spatial patches. This observation stands in stark contrast to the suggestions of Raghu et al. (2021) that ViTs possess strong localization throughout the entire network, and their further hypothesis that the addition of global pooling is required for mixing tokens at the end of the network.

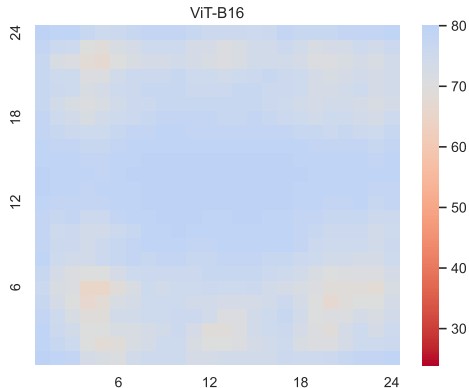

Figure 6: Heat map of classification accuracy on the validation set when we apply the classification head trained to classify images on the top of the CLS token to the other patches.

We conclude by noting that the information structure of a ViT is remarkably similar to a CNN, in the sense that the information is positionally encoded and preserved until the final layer. Furthermore, the final layer in ViTs appears to behave as a learned global pooling operation that aggregates information from all patches, which is similar to its explicit average-pooling counterpart in CNNs.

## 5 COMPARISON OF VITS AND CNNS

As extensive work has been done to understand the workings of convolutional networks, including similar feature visualization and image reconstruction techniques to those used here, we may be able to learn more about ViT behavior via direct comparison to CNNs. An important observation is that in CNNs, early layers recognize color, edges, and texture, while deeper layers pick out increasingly complex structures eventually leading to entire objects (Olah et al., 2017). Visualization of features from different layers in a ViT, such as those in Figures 1 and 7, reveal that ViTs exhibit this kind of progressive specialization as well.

On the other hand, we observe that there are also important differences between the ways CNNs and ViTs recognize images. In particular, we examine the reliance of ViTs and CNNs on background and

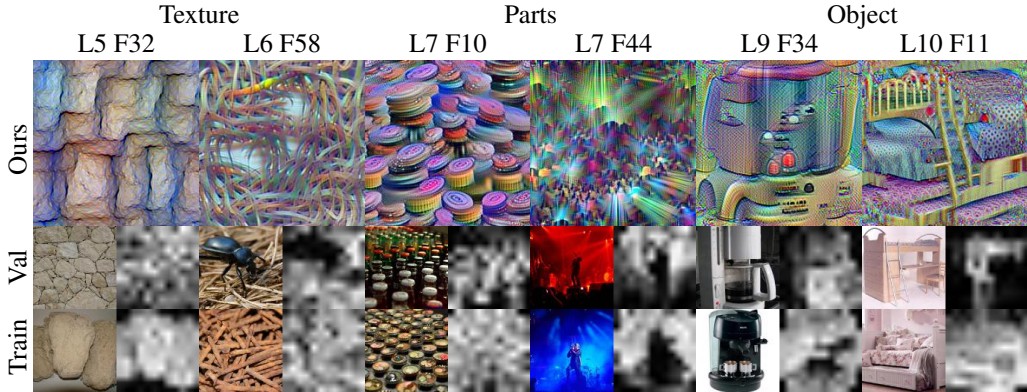

Figure 7: **Complexity of features vs depth in ViT B-32.** Visualizations suggest that ViTs are similar to CNNs in that they show a feature progression from textures to parts to objects as we progress from shallow to deep features.

foreground image features using the bounding boxes provided by ImageNet Deng et al. (2009). We filter the ImageNet-1k training images and only use those which are accompanied by bounding boxes. If several objects are present in an image, we only take the bounding boxes corresponding to the true class label and ignore the additional bounding boxes. Figure 8(b) shows an example of an image and variants in which the background and foreground, respectively, are masked.

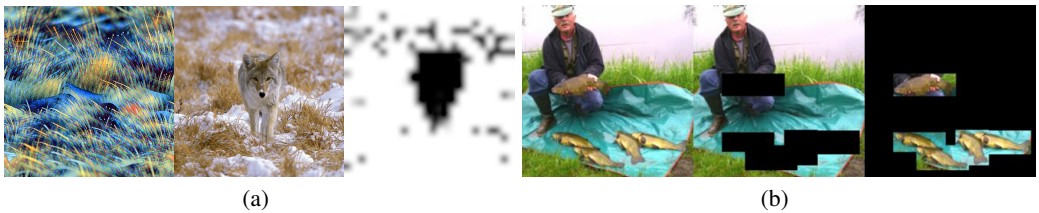

(a)                                                      (b)

Figure 8: *(a)*: **ViT-B16 detects background features**. *Left:* Image optimized to maximally activate a feature from layer 6. *Center:* Corresponding maximally activating example from ImageNet. *Right:* The image's patch-wise activation map. *(b)*: **An example of an original image and masked-out foreground and background.**

Figure 8(a) displays an example of ViTs' ability to detect background information present in the ImageNet dataset. This particular feature appears responsible for recognizing the pairing of grass and snow. The rightmost panel indicates that this feature is solely activated by the background, and not at all by the patches of the image containing parts of the wolf.

To quantitatively assess each architecture's dependence on different parts of the image on the dataset level, we mask out the foreground or background on a set of evaluation images using the aforementioned ImageNet bounding boxes, and we measure the resulting change in top-5 accuracy. These tests are performed across a number of pretrained ViT models, and we compared to a set of common CNNs in Table 2. Further results can be found in Table 3.

We observe that ViTs are significantly better than CNNs at using the background information in an image to identify the correct class. At the same time, ViTs also suffer noticeably less from the removal of the background, and thus seem to depend less on the background information to make their classification. A possible, and likely, confounding variable here is the imperfect separation of the background from the foreground in the ImageNet bounding box data set. A rectangle containing the wolf in Figure 8(a), for example, would also contain a small amount of the grass and snow at the wolf's feet. However, the foreground is typically contained entirely in a bounding box, so masking out the bounding box interiors is highly effective at removing the foreground. Because ViTs are better equipped to make sense of background information, the leaked background may be useful

Table 2: **ViTs more effectively correlate background information with correct class.** Both foreground and background data are normalized by full image top-5 accuracy.

| Normalized Top-5 ImageNet Accuracy | | | |
|---|---|---|---|
| Architecture | Full Image | Foreground | Background |
| ViT-B32 | 98.44 | 93.91 | 28.10 |
| ViT-L16 | 99.57 | 96.18 | 33.69 |
| ViT-L32 | 99.32 | 93.89 | 31.07 |
| ViT-B16 | 99.22 | 95.64 | 31.59 |
| DeiT-B16 | 99.86 | 94.98 | 38.29 |
| ConViT-B | 99.78 | 94.89 | 37.09 |
| Swin-L4 | 99.67 | **97.04** | **44.50** |
| EfficientNetB5 | 99.57 | 92.16 | 22.29 |
| EfficientNetB6 | 99.29 | 92.52 | 23.05 |
| EfficientNetB7 | 99.42 | 93.23 | 23.28 |
| ResNet-50 | 98.00 | 89.69 | 18.69 |
| ResNet-152 | 98.85 | 90.74 | 19.68 |
| MobileNetv2 | 96.09 | 86.84 | 15.94 |
| DenseNet121 | 96.55 | 89.58 | 17.53 |

for maintaining superior performance. Nonetheless, these results suggest that ViTs consistently outperform CNNs when information, either foreground or background, is missing.

Next, we study the role of texture in ViT predictions. To this end, we filter out high-frequency components from ImageNet test images via low-pass filtering. While the predictions of ResNets suffer greatly when high-frequency texture information is removed from their inputs, ViTs are seemingly resilient. See Figure 15 for the decay in accuracy of ViT and ResNet models as textural information is removed.

## 6 ViTs with Language Model Supervision

Recently, ViTs have been used as a backbone to develop image classifiers trained with natural language supervision and contrastive learning techniques (Radford et al., 2021). These *CLIP* models are state-of-the-art in transfer learning to unseen datasets. The zero-shot ImageNet accuracy of these models is even competitive with traditionally trained ResNet-50 competitors. We compare the feature visualizations for ViT models with and without CLIP training to study the effect of natural language supervision on the behavior of the transformer-based backbone.

The training objective for CLIP models consists of matching the correct caption from a list of options with an input image (in feature space). Intuitively, this procedure would require the network to extract features not only suitable for detecting nouns (e.g. simple class labels like 'bird'), but also modifying phrases like prepositions and epithets. Indeed, we observe several such features that are not present in ViTs trained solely as image classifiers.

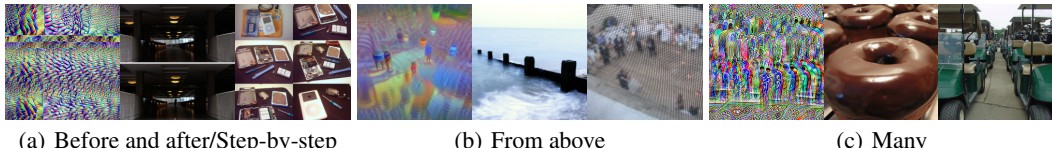

(a) Before and after/Step-by-step     (b) From above     (c) Many

Figure 9: *Left:* Feature optimization shows sharp boundaries, and maximally activating ImageNet examples contain distinct, adjacent images. *Middle:* Feature optimization and maximally activating ImageNet photos all show images from an elevated vantage point. *Right:* Feature optimization shows a crowd of people, but maximally activating images indicate that the repetition of objects is more relevant than the type of object.

Figure 9(a) shows the image optimized to maximally activate a feature of a ViT CLIP model alongside its two highest activating examples from the ImageNet dataset. The fact that all three images share

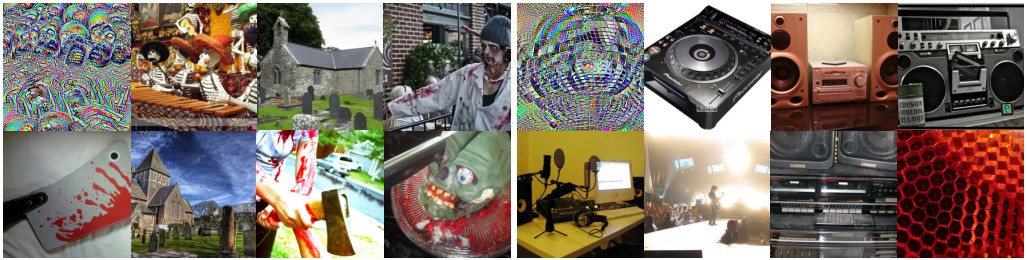

(a) Category of morbidity            (b) Category of music

Figure 10: **Features from ViT trained with CLIP that relates to the category of morbidity and music.** *Top-left image in each category:* Image optimized to maximally activate a feature from layer 10. *Rest:* Seven of the ten ImageNet images that most activate the feature.

sharp boundaries indicates this feature might be responsible for detecting caption texts relating to a progression of images. Examples could include "before and after," as in the airport images or the adjective "step-by-step" for the iPod teardown. Similarly, Figure 9(b) and 9(c) depict visualizations from features which seem to detect the preposition "from above", and adjectives relating to a multitude of the same object, respectively.

The presence of features that represent conceptual categories is another consequence of CLIP training. Unlike ViTs trained as classifiers, in which features detect single objects or common background information, CLIP-trained ViTs produce features in deeper layers activated by objects in clearly discernible conceptual categories. For example, the top left panel of Figure 10(a) shows a feature activated by what resembles skulls alongside tombstones. The corresponding seven highly activating images from the dataset include other distinct objects such as bloody weapons, zombies, and skeletons. From a strictly visual point of view, these classes have very dissimilar attributes, indicating this feature might be responsible for detecting components of an image relating broadly to morbidity. In Figure 10(b), we see that the top leftmost panel shows a disco ball, and the corresponding images from the dataset contain boomboxes, speakers, a record player, audio recording equipment, and a performer. Again, these are visually distinct classes, yet they are all united by the concept of music.

Given that the space of possible captions for images is substantially larger than the mere one thousand classes in the ImageNet dataset, high performing CLIP models understandably require higher level organization for the objects they recognize. Moreover, the CLIP dataset is scraped from the internet, where captions are often more descriptive than simple class labels.

## 7 DISCUSSION

In order to dissect the inner workings of vision transformers, we introduce a framework for optimization-based feature visualization. We then identify which components of a ViT are most amenable to producing interpretable images, finding that the high-dimensional inner projection of the feed-forward layer is suitable while the key, query, and value features of self-attention are not.

Applying this framework to said features, we observe that ViTs preserve spatial information of the patches even for individual channels across all layers with the exception of the last layer, indicating that the networks learn spatial relationships from scratch. We further show that the sudden disappearance of localization information in the last attention layer results from a learned token mixing behavior that resembles average pooling.

In comparing CNNs and ViTs, we find that ViTs make better use of background information and are able to make vastly superior predictions relative to CNNs when exposed only to image backgrounds despite the seemingly counter-intuitive property that ViTs are not as sensitive as CNNs to the loss of high-frequency information, which one might expect to be critical for making effective use of background. We also conclude that the two architectures share a common property whereby earlier layers learn textural attributes, whereas deeper layers learn high level object features or abstract concepts. Finally, we show that ViTs trained with language model supervision learn more semantic and conceptual features, rather than object-specific visual features as is typical of classifiers.

## REPRODUCIBILITY STATEMENT

We make our code repository available at: `https://github.com/anonymous2023iclr/ViTVis`

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

## A  LIMITATIONS

The paper highlights that the technique we utilize to create visual representations is ineffective for features with low dimensionality in the network. However, we have managed to address this issue by discovering that visualizing features with high dimensionality produces satisfactory results.

## B  FAILED EXAMPLES

Figure 11 shows few examples of our visualization method failing when applied on low dimensional spaces. We attribute this to entanglement of more than 768 features when represented by vectors of size 768. We note that, due to skip connections, activation in previous layers can cause activation in the next layer for the same feature, consequently, the visualizations of the same features in different layers can share visual similarities.

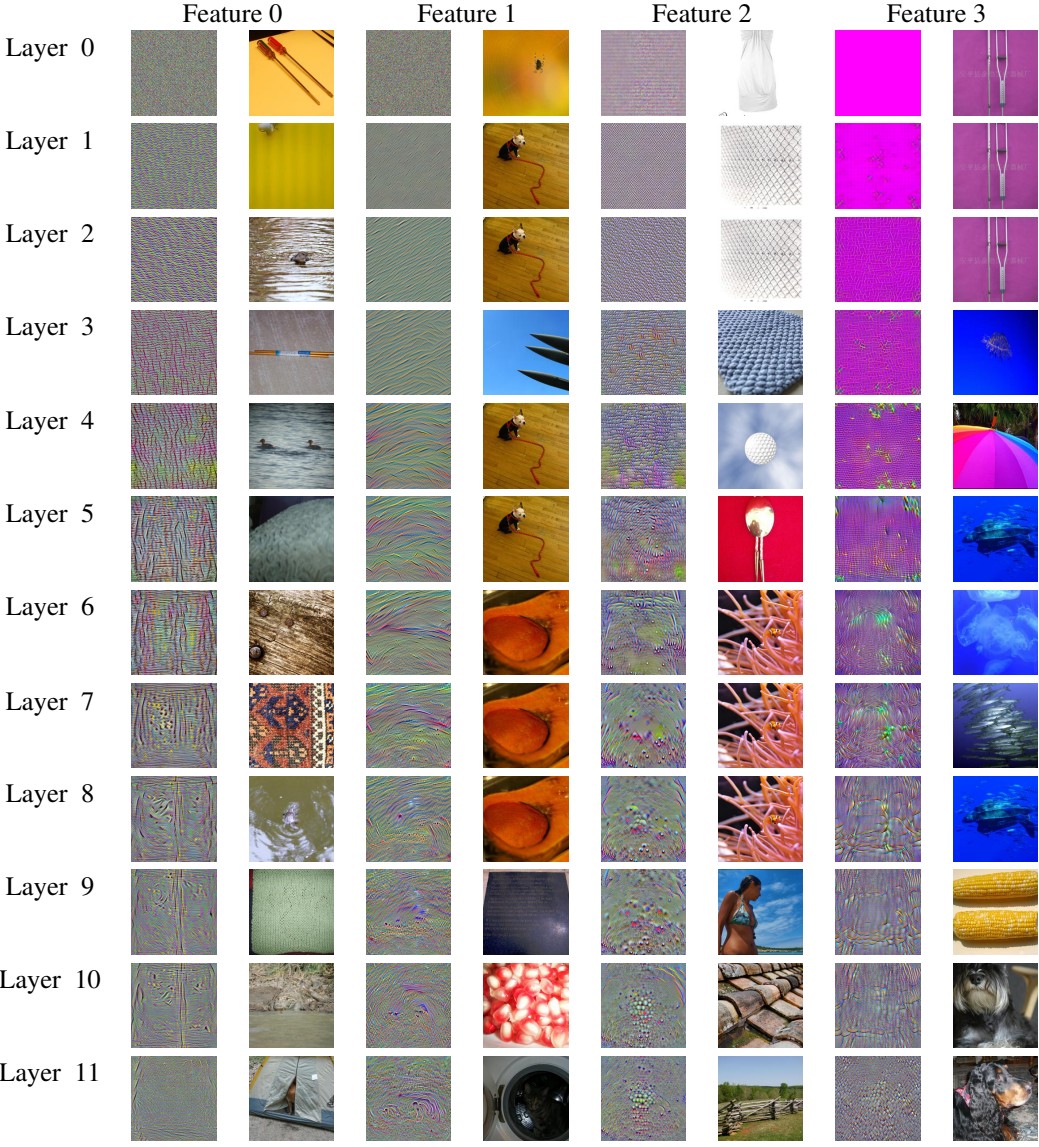

Figure 11: **Some examples of failed visualizations on the input of the attention layers**. Same visualization technique fails when applied on low dimensional (e.g. on key, query, value, etc) spaces. We believe that the visualization shows roughly meaningful and interpretable visualizations in early layers, since there are not many different features to be embedded. However, in deeper layers the features are entangled, so it is more difficult to visualize them. For every example, the picture on the left shows the results of optimization and the picture on the right shows the most activating image from ImageNet1k validation set.

## C   EXPERIMENTAL SETUP AND HYPERPARAMETERS

As mentioned before, we ensemble augmentations to the input. More specifically, we use $\mathcal{A}$ is $GS(CS(Jitter(x)))$ as our augmentation. The bound for $Jitter$ is $(-32, 32)$ for both directions vertical and horizontal. The hyper parameters for $CS$ are always $mean = 1.0$ and $std = 1.0$ in all of the experiments. For $GS$, the $mean$ is always $0$, however, for the $std$ we have a linear scheduling, where at the beginning of the optimization the $std = 0.5$ and at the end of the optimization $std = 0.0$. We use a batch-size $n = 8$ for all of our experiments. We use ADAM as our choice of optimizer with $\beta = (0.5, 0.99)$. Optimization is done in $400$ steps and at every step, we re-sample the augmentations $Jitter$, $GS$ and $CS$. We also use a CosineAnealing learning rate scheduling, starting from $lr = 0.1$ at the beginning and $l = 0$ the end. The hyper-parameter used for total variation $\lambda_{tv} = 0.00005$.

For all ViT experiments, we use the pretrained models from https://github.com/lukemelas/PyTorch-Pretrained-ViT. For clip models, we use pretrained models by Wightman (2019). The rest of models we use are from https://github.com/pytorch/vision.

For all of our experiments, we use GeForce RTX 2080 Ti GPUs with 12GB of memory. All inferences on ImageNet are done under 20 minutes on validation set and under 1 hour on training set using only 1 GPU. All visualization experiments take at most 90 seconds to complete.

## D   MODELS

In our experiments, we use publicly available pre-trained models from various sources. The following tables list the models used from each source, along with references to where they are introduced in the literature.

| Name | Paper |
|---|---|
| B_16_imagenet1k | Dosovitskiy et al. (2021) |
| B_32_imagenet1k | Dosovitskiy et al. (2021) |

Figure 12: Pre-trained models used from : https://github.com/lukemelas/PyTorch-Pretrained-ViT.

| Name | Paper |
|---|---|
| deit_base_patch16_224 | Touvron et al. (2021b) |
| deit_base_distilled_patch16_384 | Touvron et al. (2021b) |
| deit_base_patch16_384 | Touvron et al. (2021b) |
| deit_tiny_distilled_patch16_224 | Touvron et al. (2021b) |
| deit_small_distilled_patch16_224 | Touvron et al. (2021b) |
| deit_base_distilled_patch16_224 | Touvron et al. (2021b) |

Figure 13: Pre-trained models from Touvron et al. (2021a) .

| Name | Paper |
|------|-------|
| coat_lite_mini | Xu et al. (2021) |
| coat_lite_small | Xu et al. (2021) |
| coat_lite_tiny | Xu et al. (2021) |
| coat_mini | Xu et al. (2021) |
| coat_tiny | Xu et al. (2021) |
| convit_base | d'Ascoli et al. (2021) |
| convit_small | d'Ascoli et al. (2021) |
| convit_tiny | d'Ascoli et al. (2021) |
| pit_b_224 | Heo et al. (2021) |
| pit_b_distilled_224 | Heo et al. (2021) |
| pit_s_224 | Heo et al. (2021) |
| pit_s_distilled_224 | Heo et al. (2021) |
| pit_ti_224 | Heo et al. (2021) |
| pit_ti_distilled_224 | Heo et al. (2021) |
| swin_base_patch4_window7_224 | Liu et al. (2021) |
| swin_base_patch4_window7_224_in22k | Liu et al. (2021) |
| swin_base_patch4_window12_384 | Liu et al. (2021) |
| swin_base_patch4_window12_384_in22k | Liu et al. (2021) |
| swin_large_patch4_window7_224 | Liu et al. (2021) |
| swin_large_patch4_window7_224_in22k | Liu et al. (2021) |
| swin_large_patch4_window12_384 | Liu et al. (2021) |
| swin_large_patch4_window12_384_in22k | Liu et al. (2021) |
| swin_small_patch4_window7_224 | Liu et al. (2021) |
| swin_tiny_patch4_window7_224 | Liu et al. (2021) |
| twins_pcpvt_base | Chu et al. (2021) |
| twins_pcpvt_large | Chu et al. (2021) |
| twins_pcpvt_small | Chu et al. (2021) |
| twins_svt_base | Chu et al. (2021) |
| twins_svt_large | Chu et al. (2021) |
| twins_svt_small | Chu et al. (2021) |

Figure 14: Pre-trained models used from: Wightman (2019)

## E    EFFECT OF LOW-PASS FILTERING

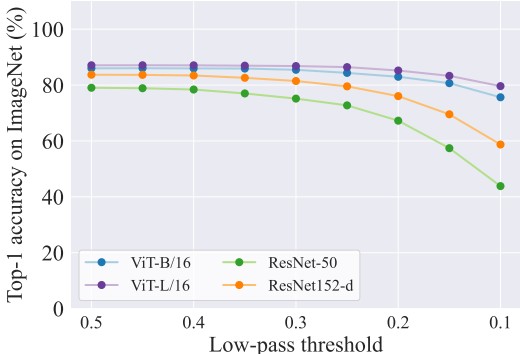

Figure 15: **Effect of low-pass filtering on top-1 ImageNet accuracy.** CNNs are *more* dependent on high frequency textural image information than ViTs.

## F    ADDITIONAL VISUALIZATIONS

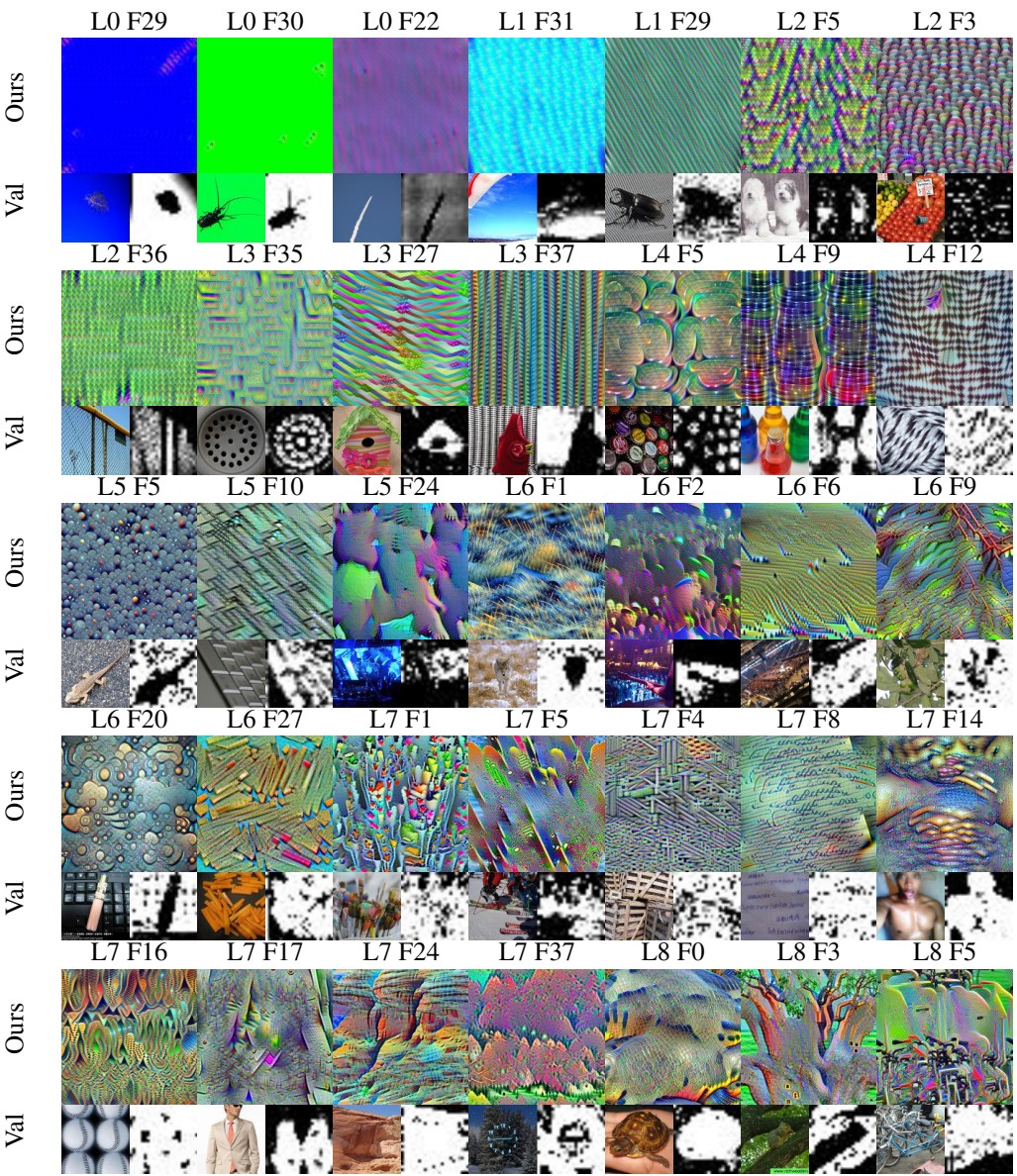

Figure 16: Visualization of ViT-base-patch16

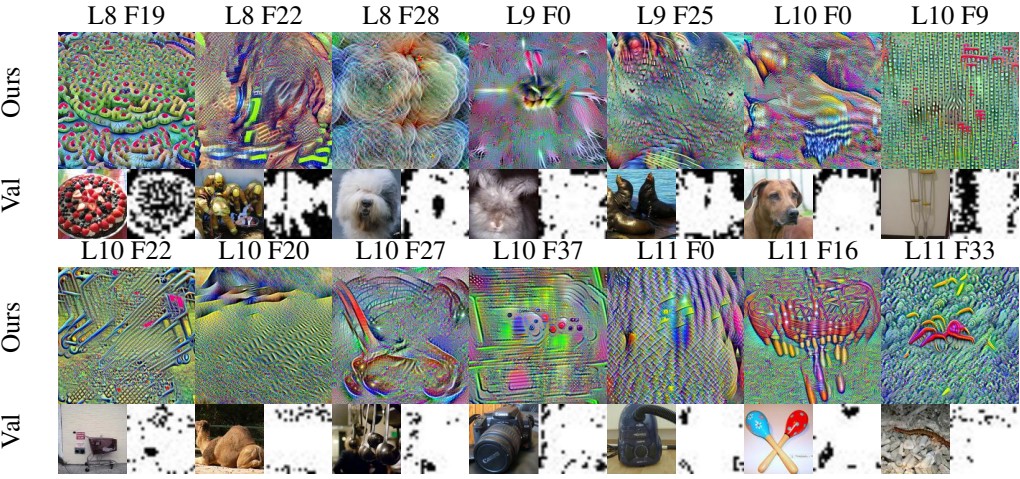

Figure 17: (Cont.) Visualization of ViT-base-patch16

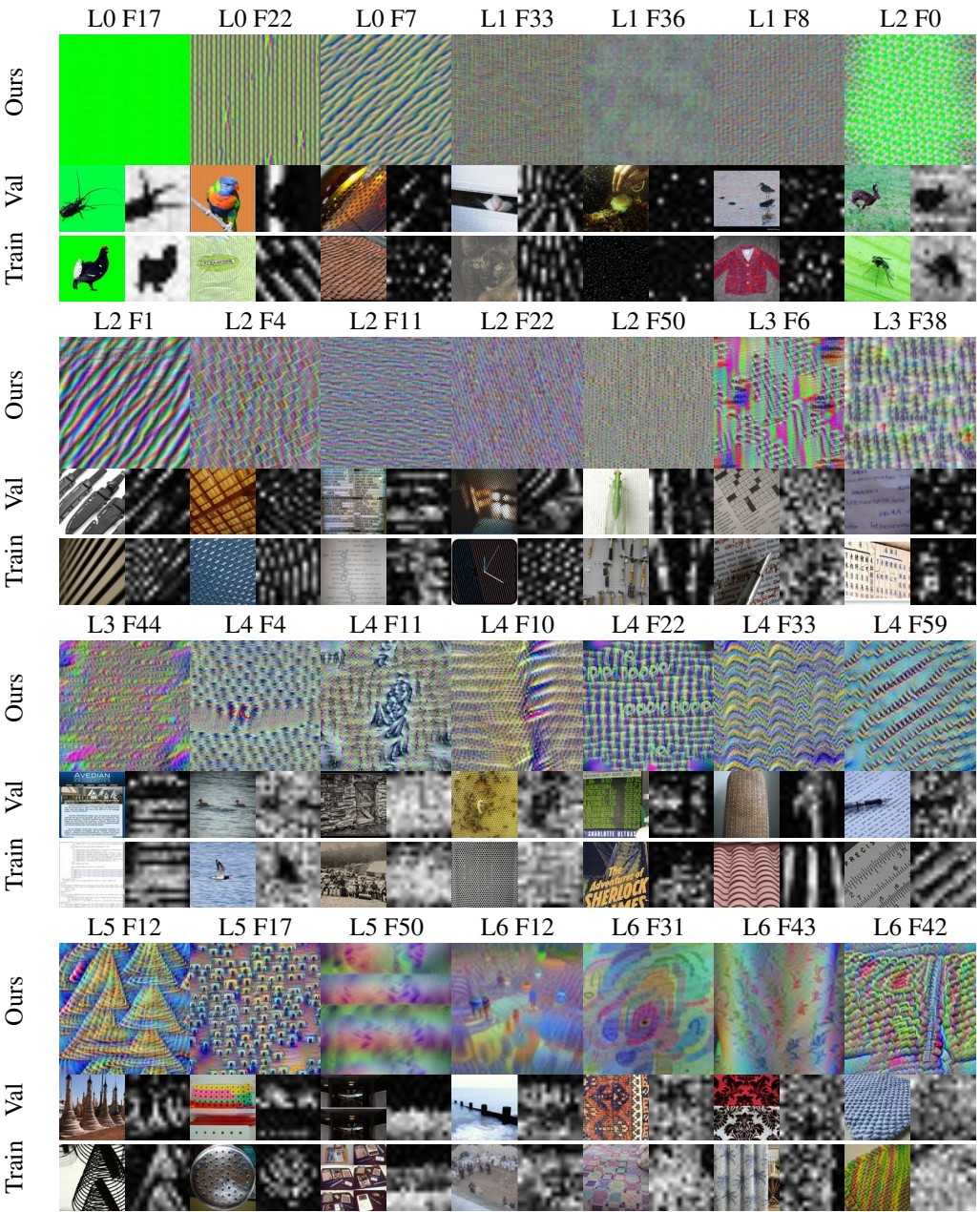

Figure 18: Visualization of a CLIP model with ViT-base-patch16 as its visual part.

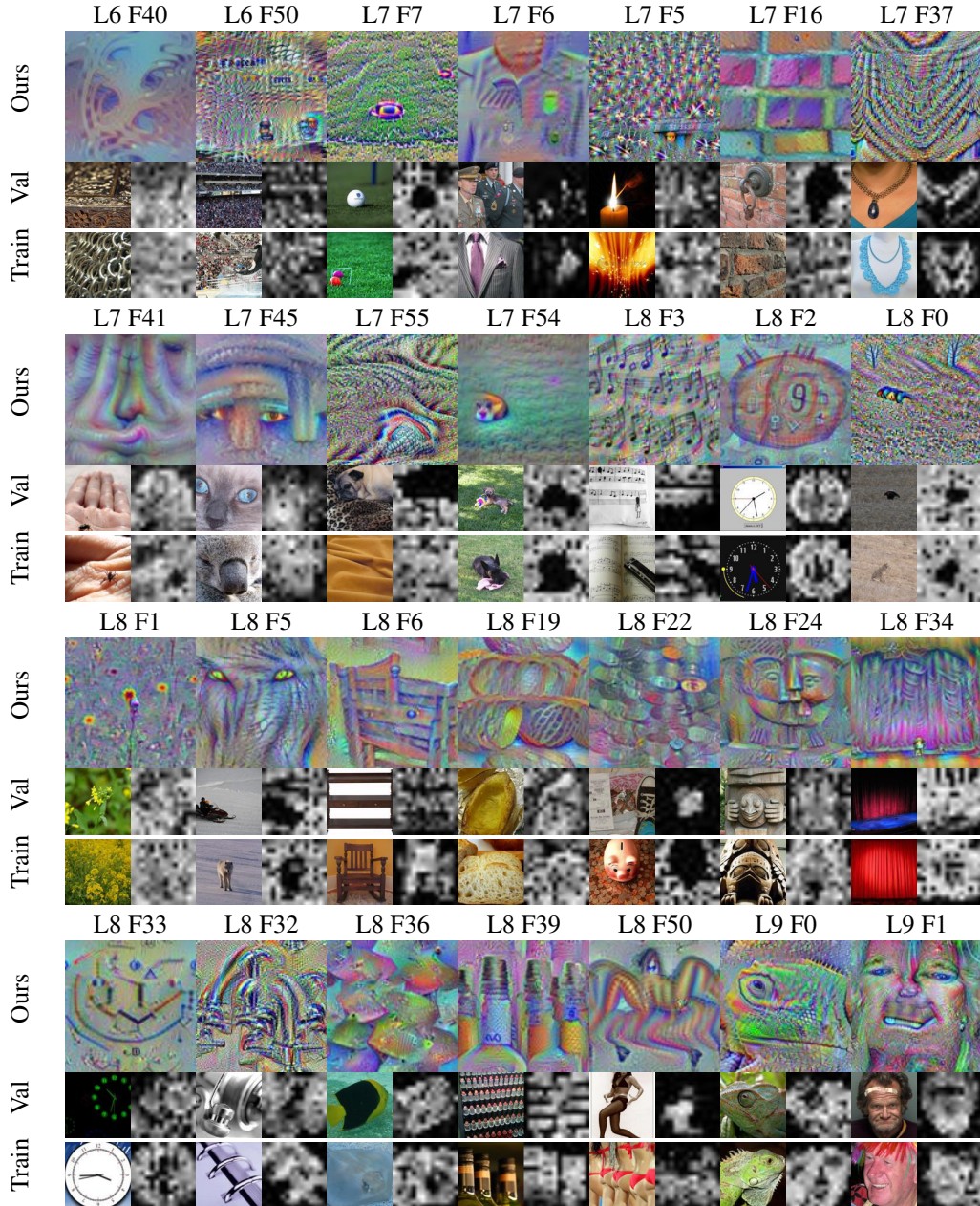

Figure 19: (Cont.) Visualization of a CLIP model with ViT-base-patch16 as its visual part.

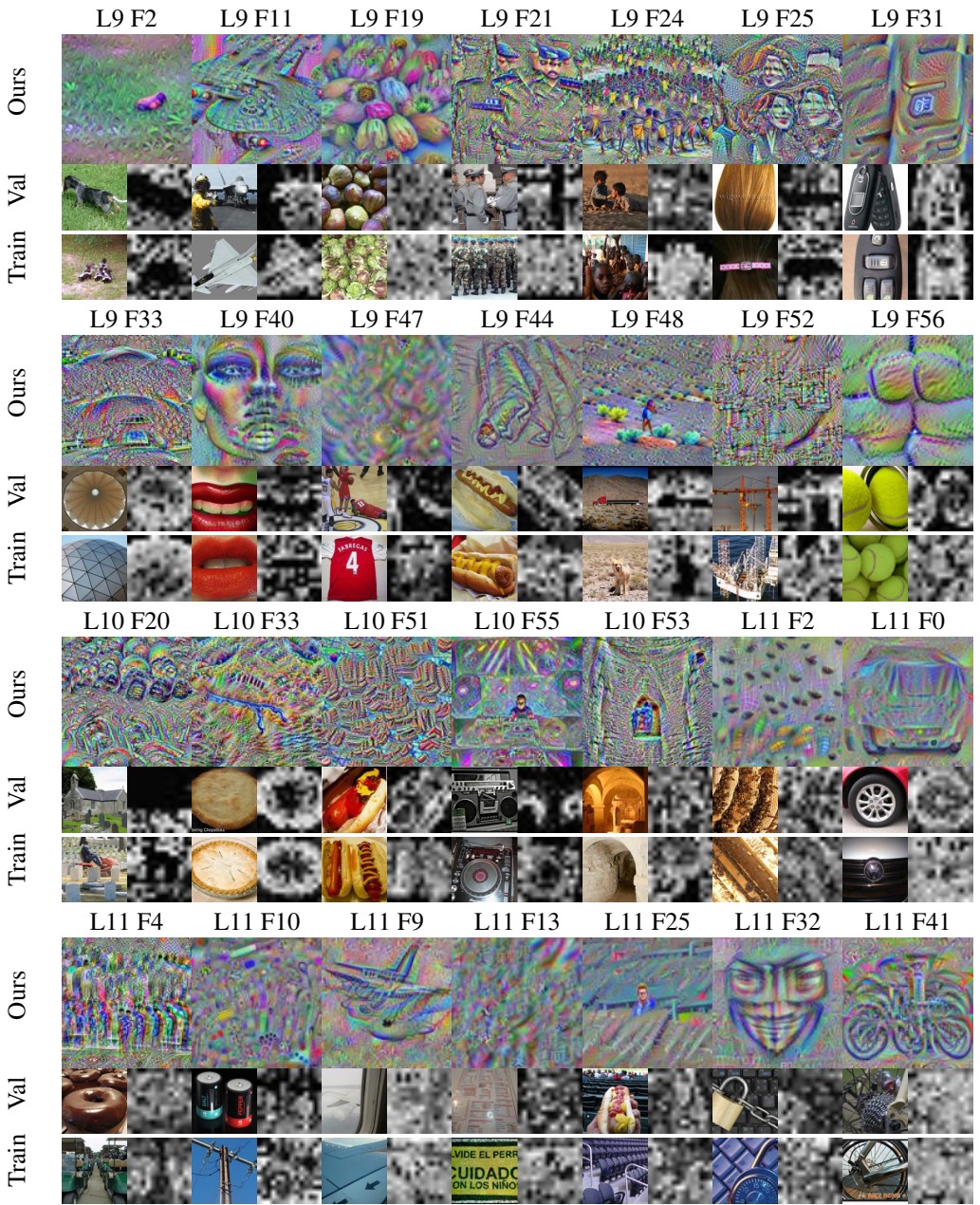

Figure 20: (Cont.) Visualization of a CLIP model with ViT-base-patch16 as its visual part.

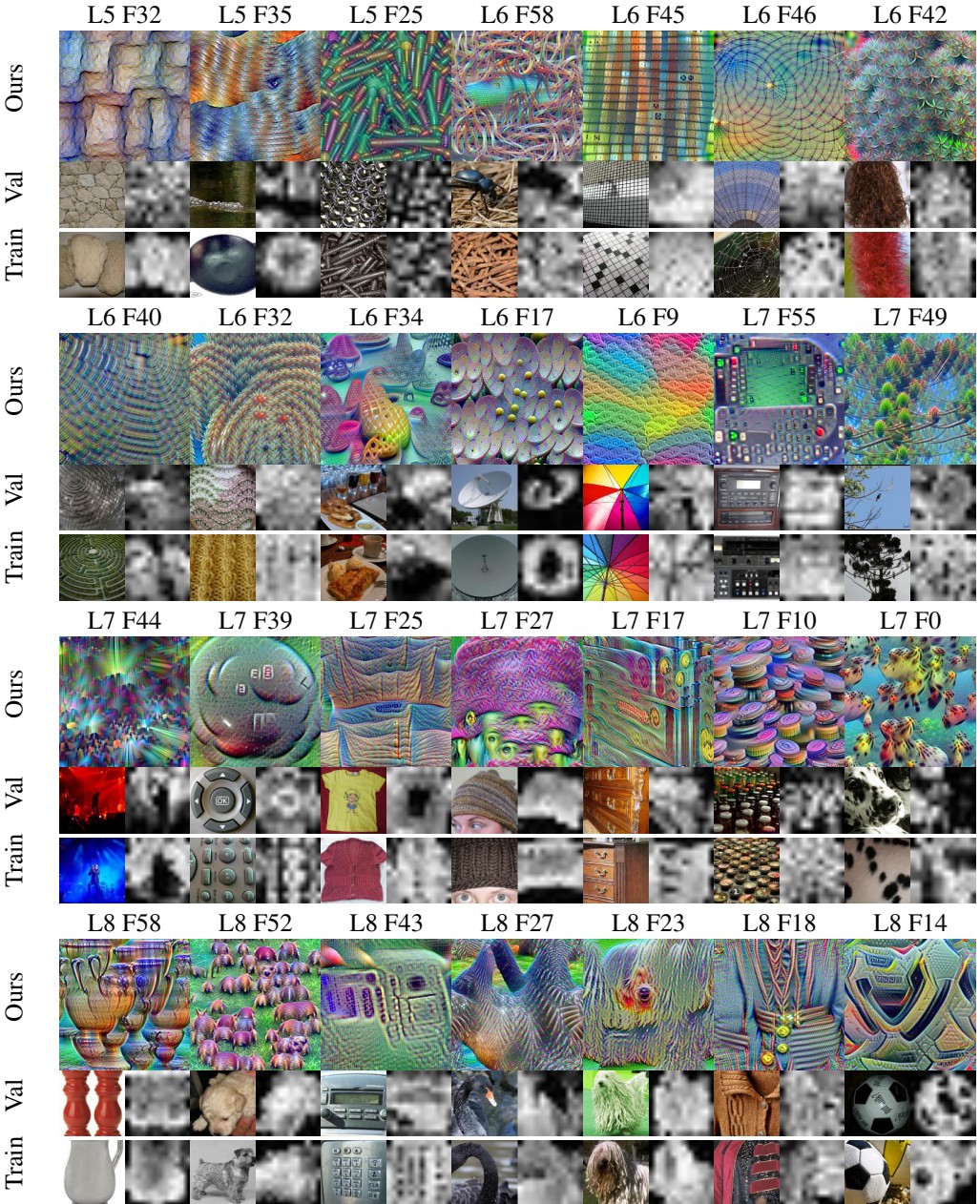

Figure 21: Visualization of ViT-base-patch32

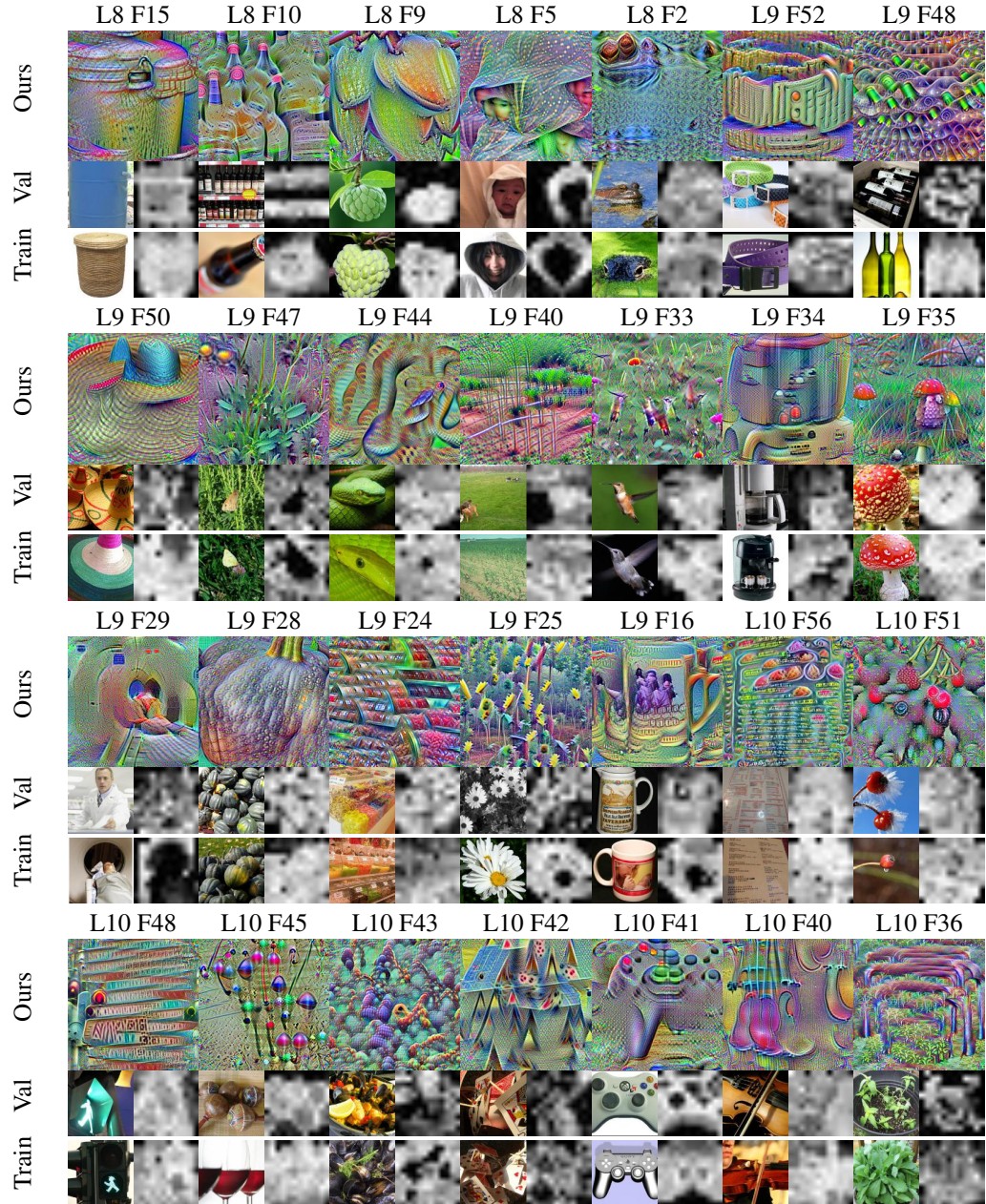

Figure 22: (Cont.) Visualization of ViT-base-patch32

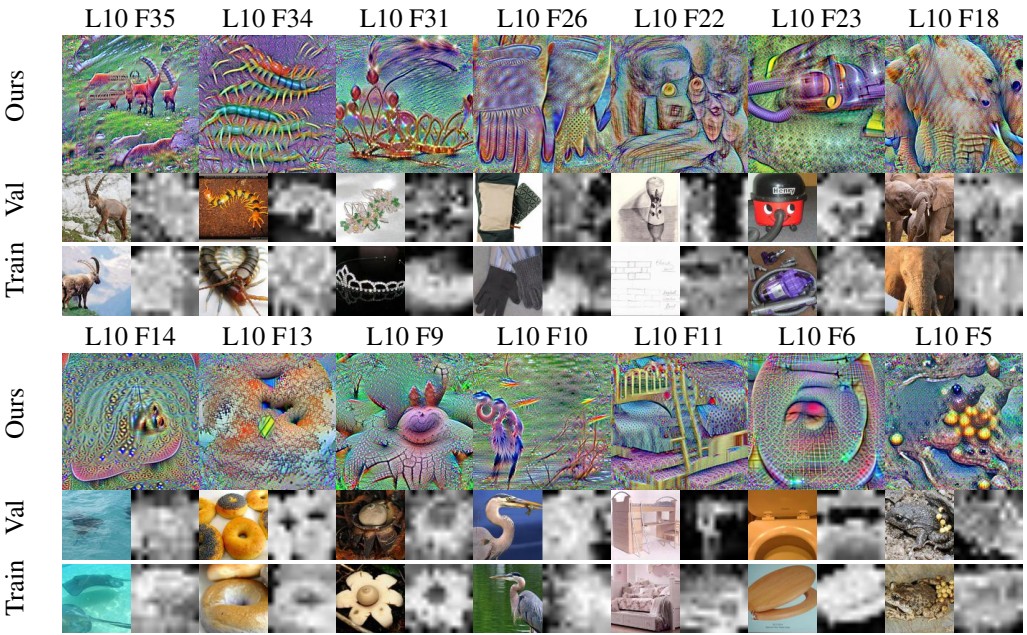

Figure 23: (Cont.) Visualization of ViT-base-patch32

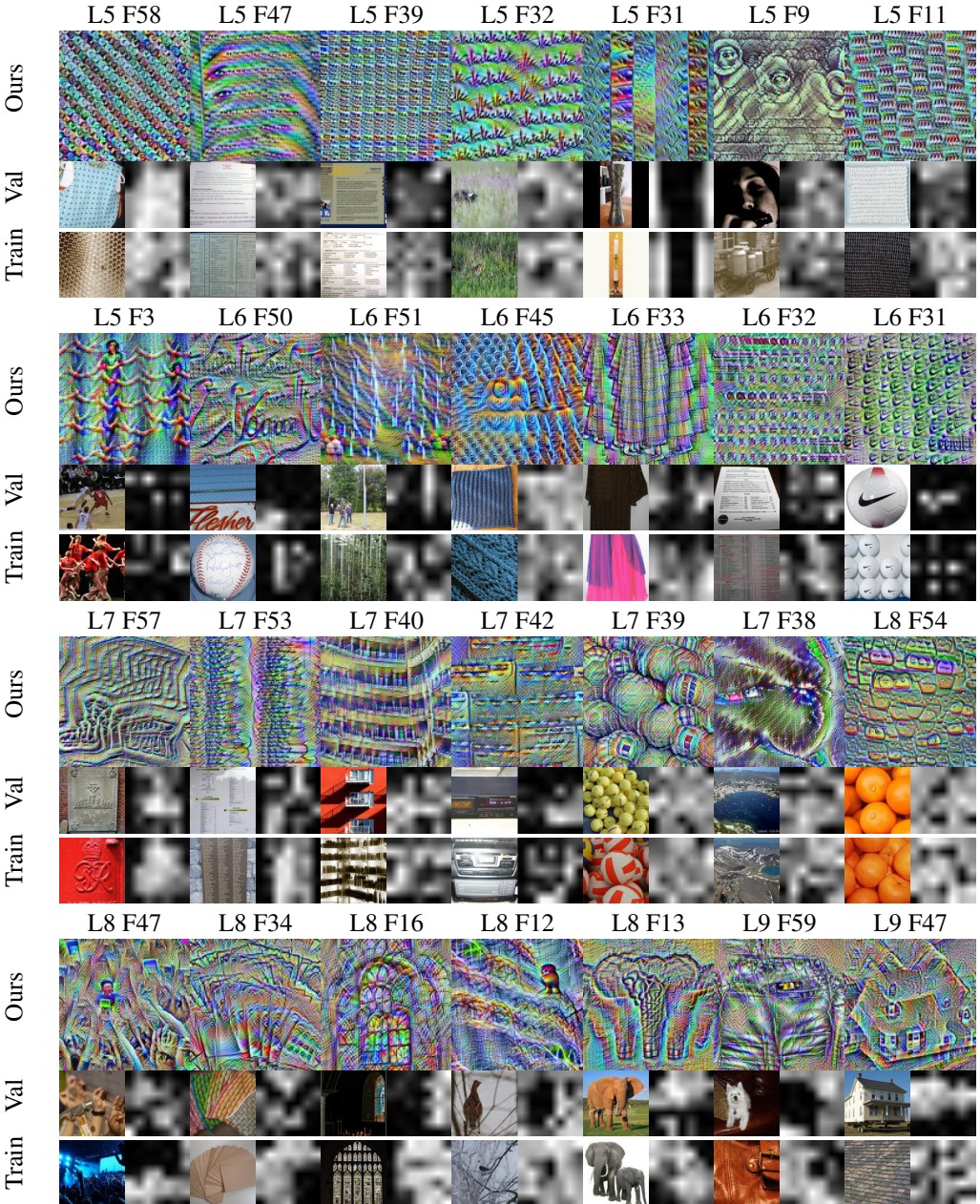

Figure 24: Visualization of a CLIP model with ViT-base-patch32 as its visual part.

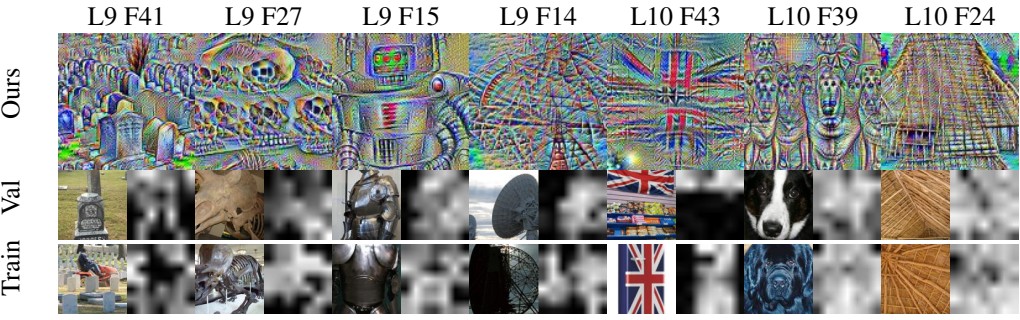

Figure 25: (Cont.) Visualization of a CLIP model with ViT-base-patch32 as its visual part.

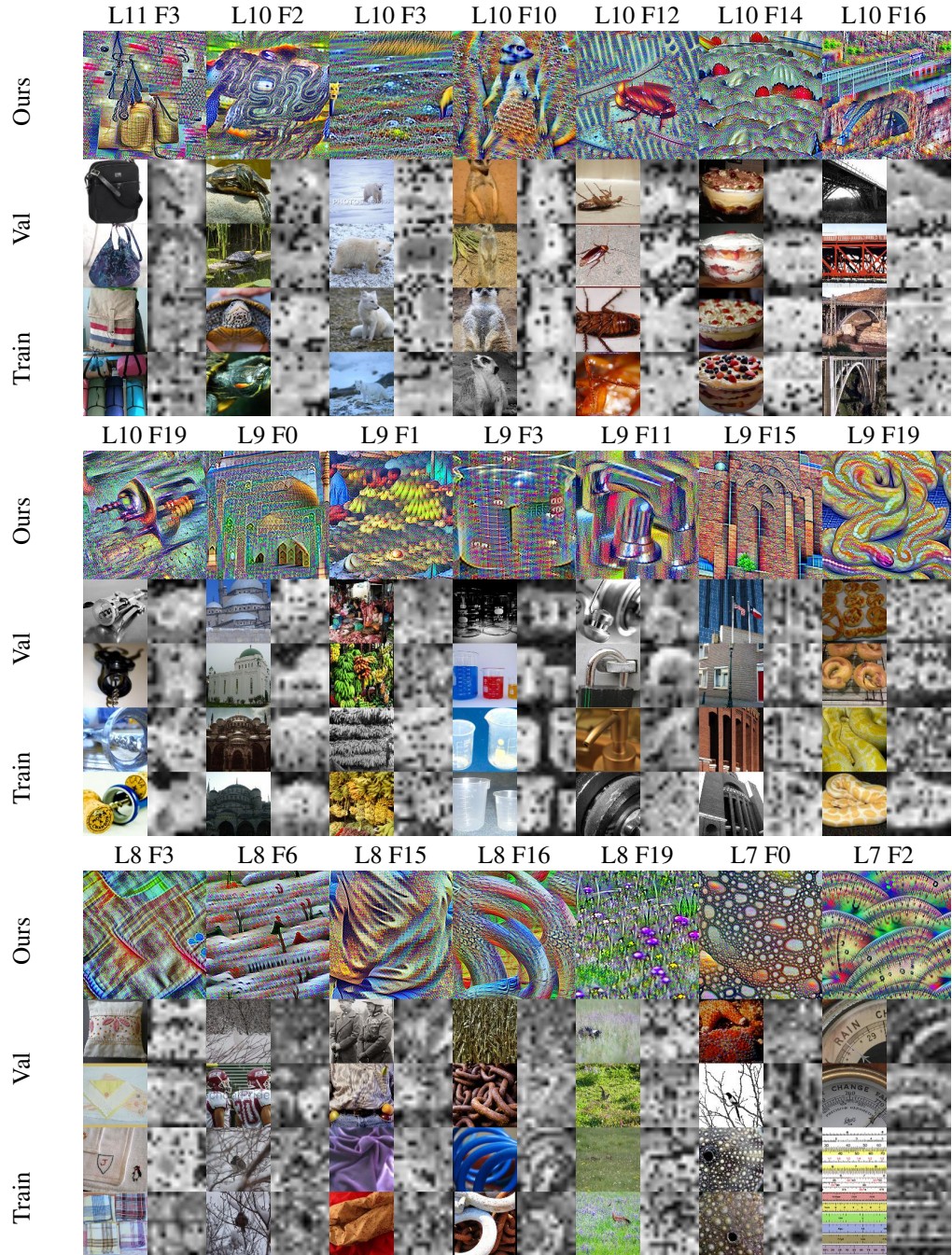

Figure 26: Visualization of features in Deit base p-16 im-224

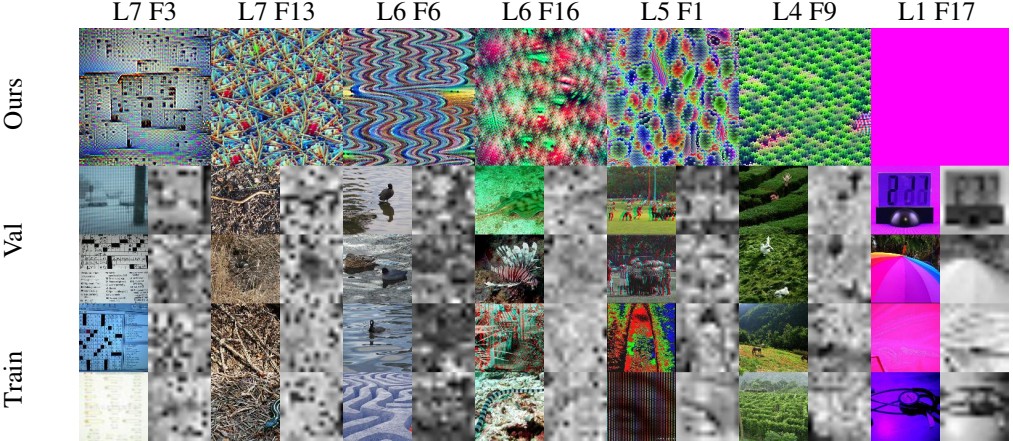

Figure 27: (Cont.) Visualization of features in Deit base p-16 im-224

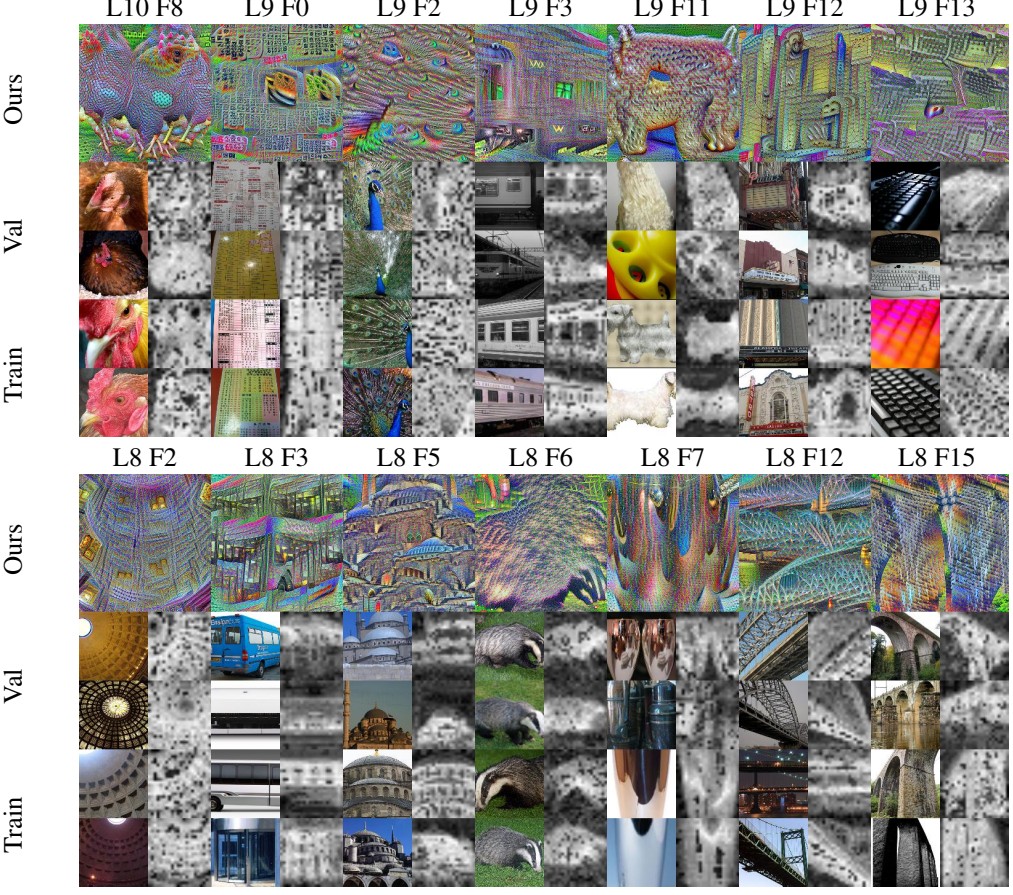

Figure 28: Visualization of features in DeiT base p-16 im-384

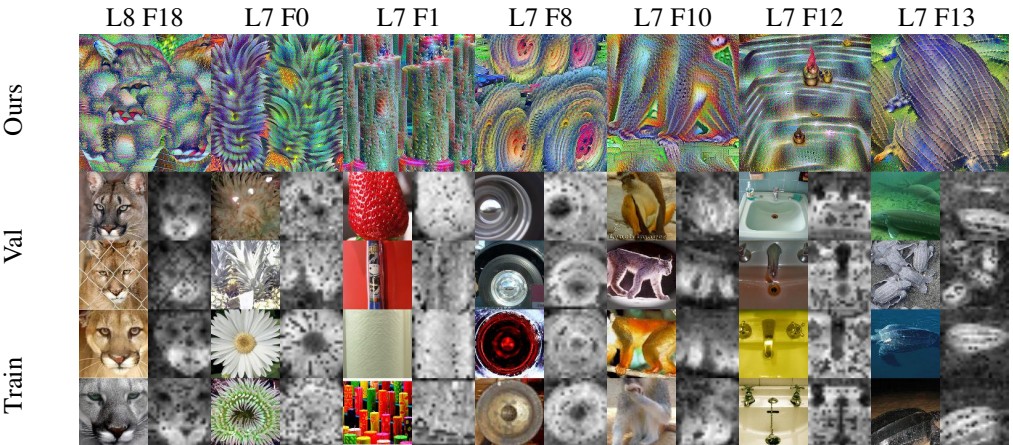

Figure 29: (Cont.) Visualization of features in DeiT base p-16 im-384

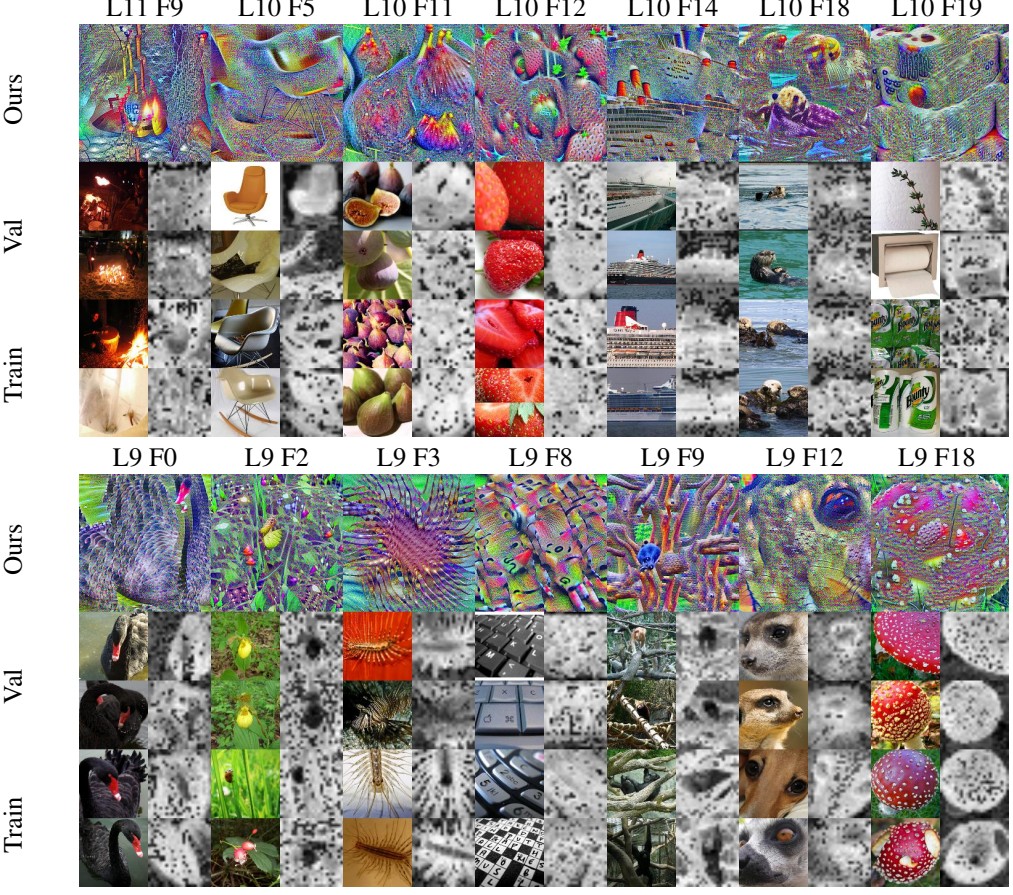

Figure 30: Visualization of features in DeiT base p-16 im-384

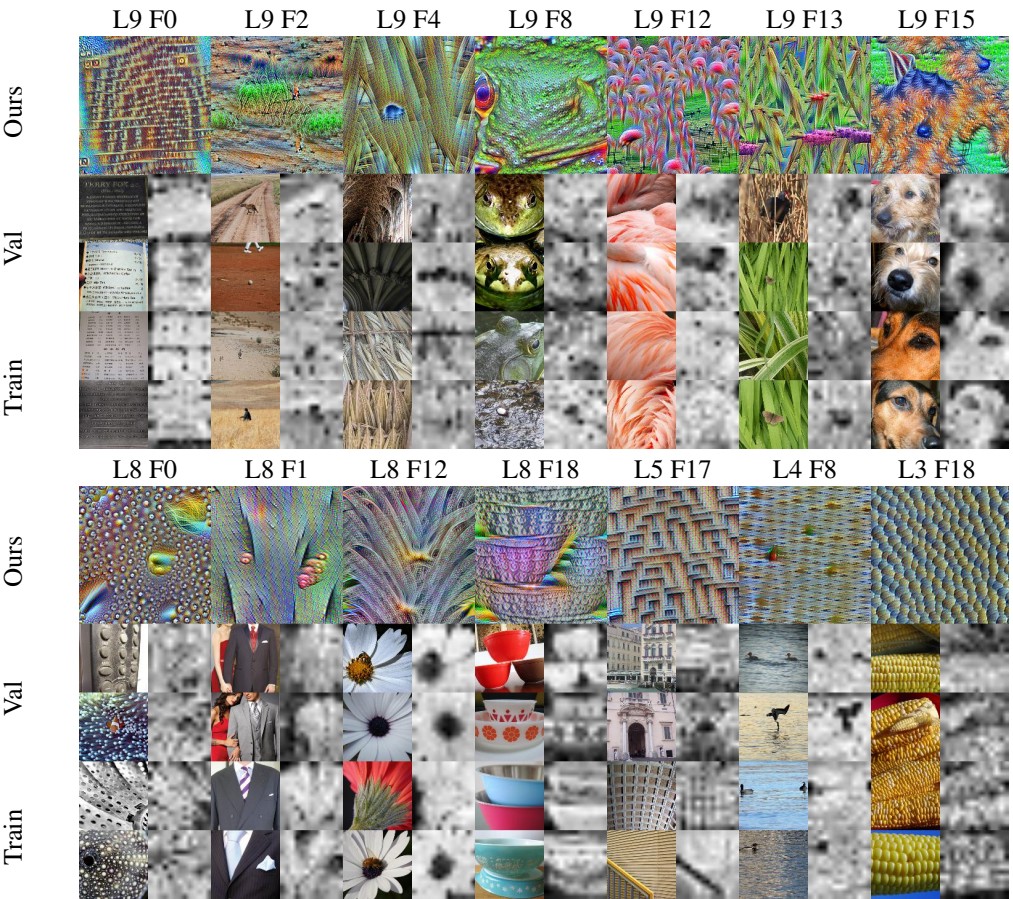

Figure 31: Visualization of features in DeiT tiny distilled p-16 im-224

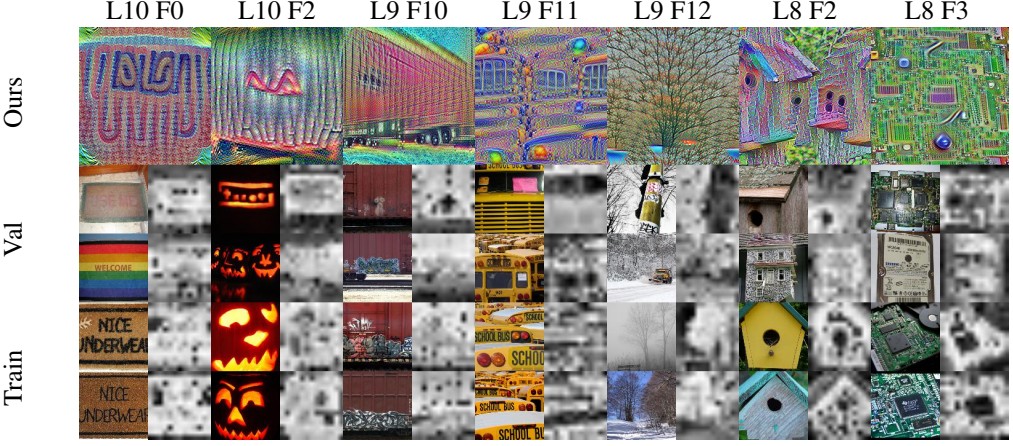

Figure 32: Visualization of features in DeiT small distilled p-16 im-224

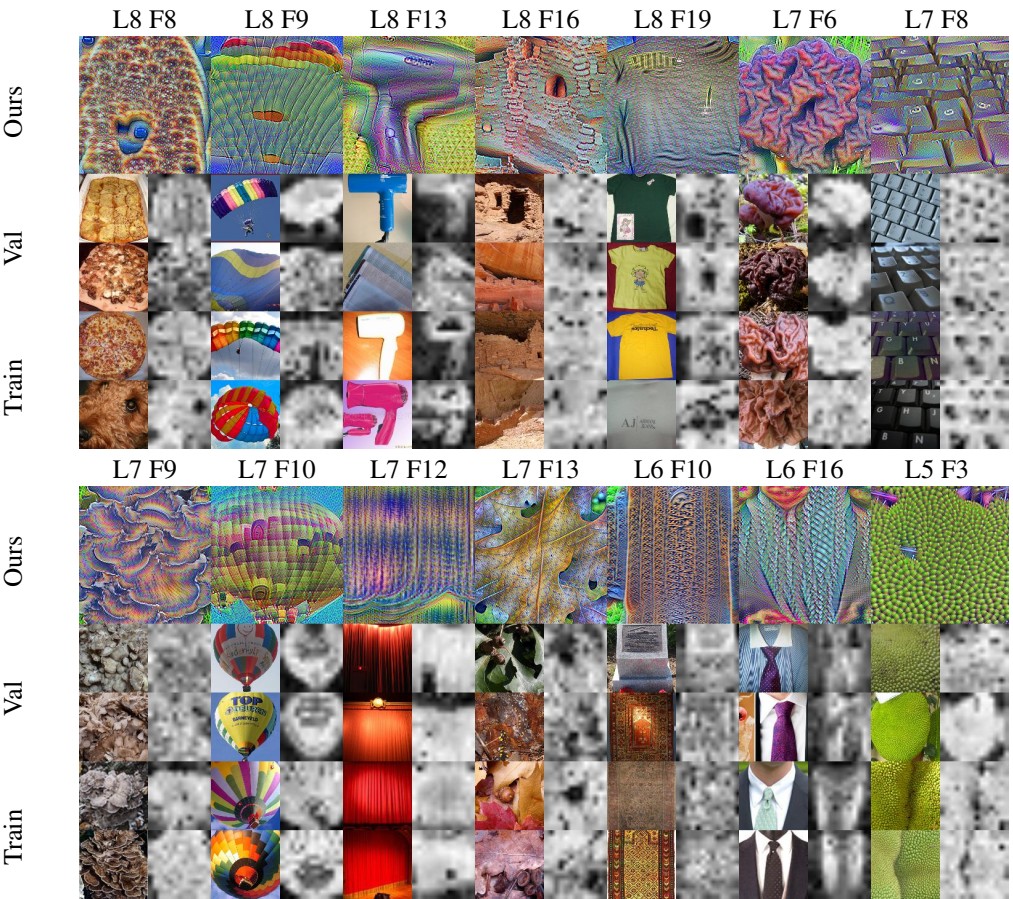

Figure 33: Visualization of features in DeiT small distilled p-16 im-224

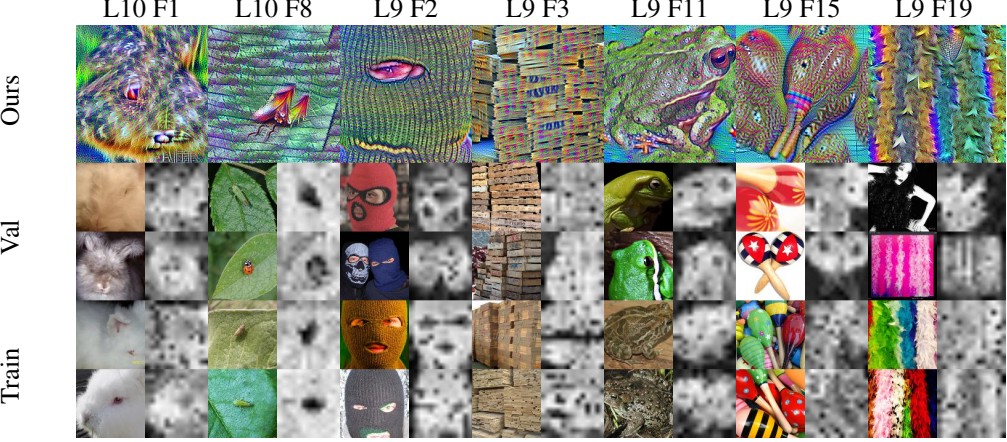

Figure 34: Visualization of features in DeiT base distilled p-16 im-224

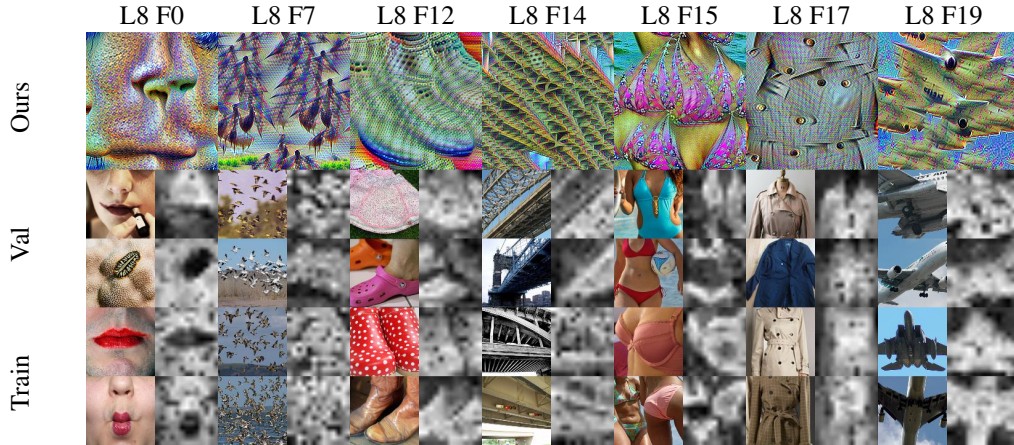

Figure 35: (Cont.) Visualization of features in DeiT base distilled p-16 im-224

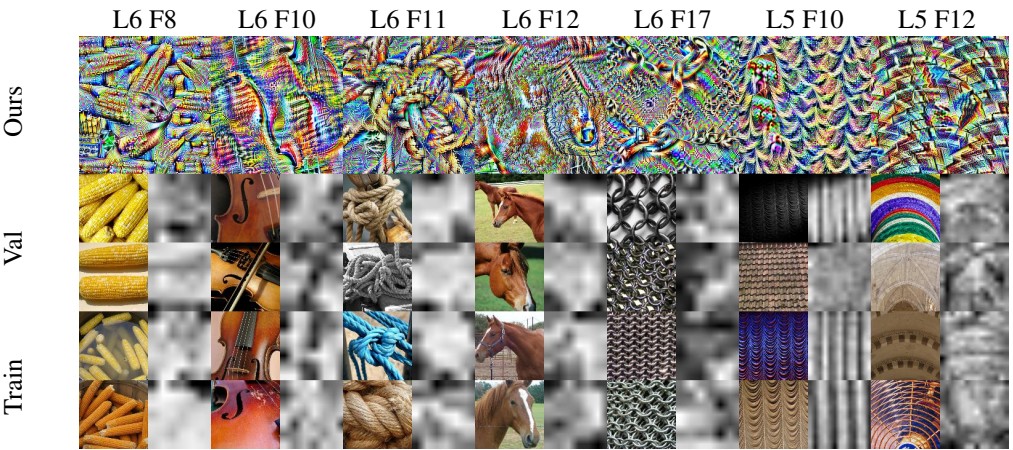

Figure 36: Visualization of features in Coat lite mini

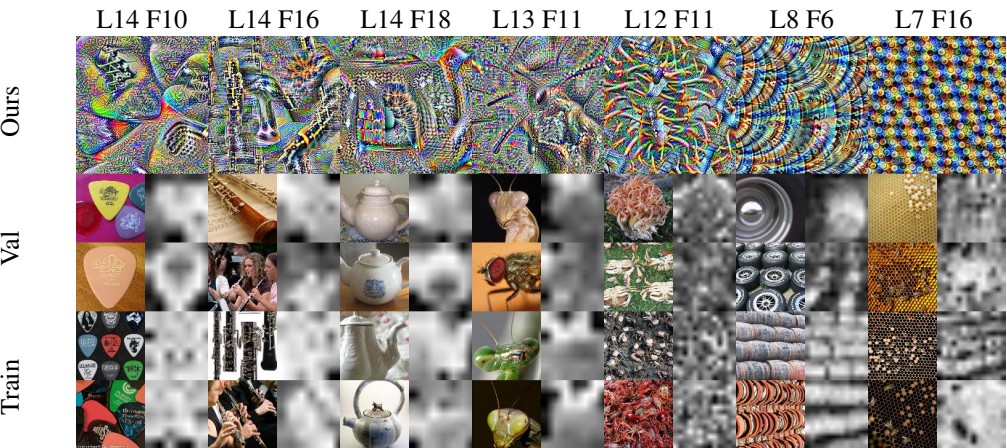

Figure 37: Visualization of features in Coat lite small

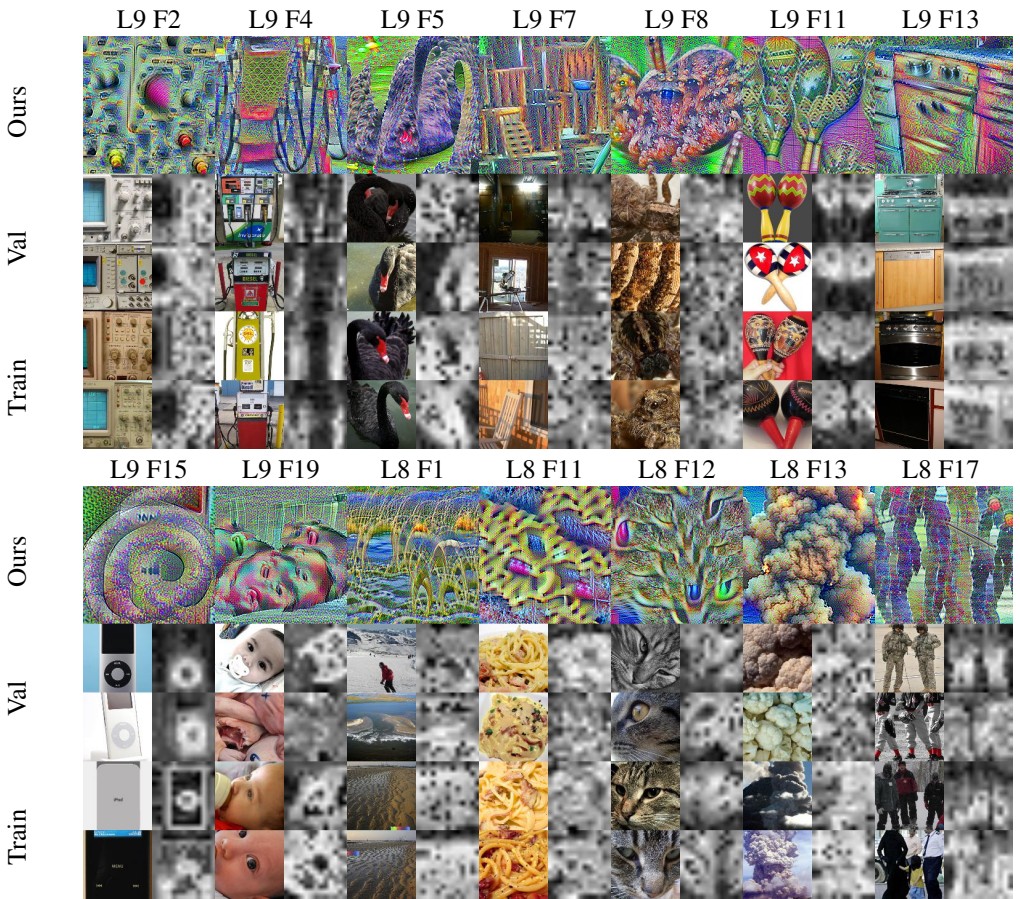

Figure 38: Visualization of features in ConViT base

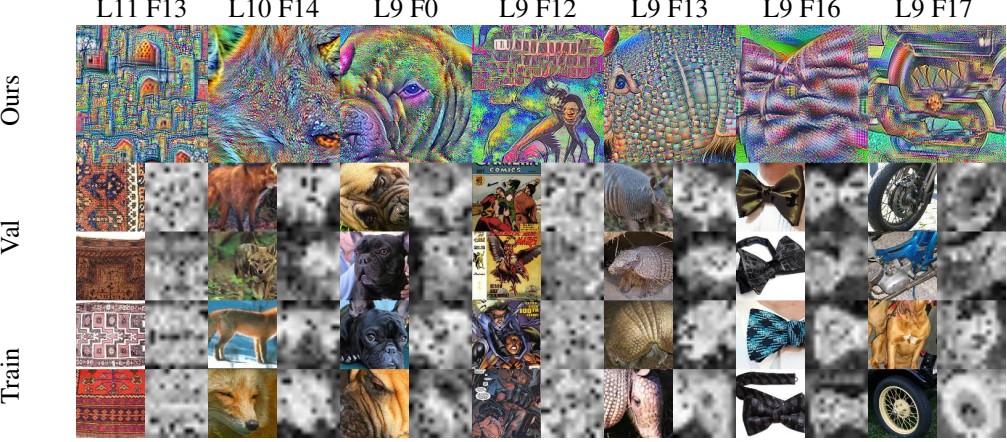

Figure 39: Visualization of features in ConViT small.

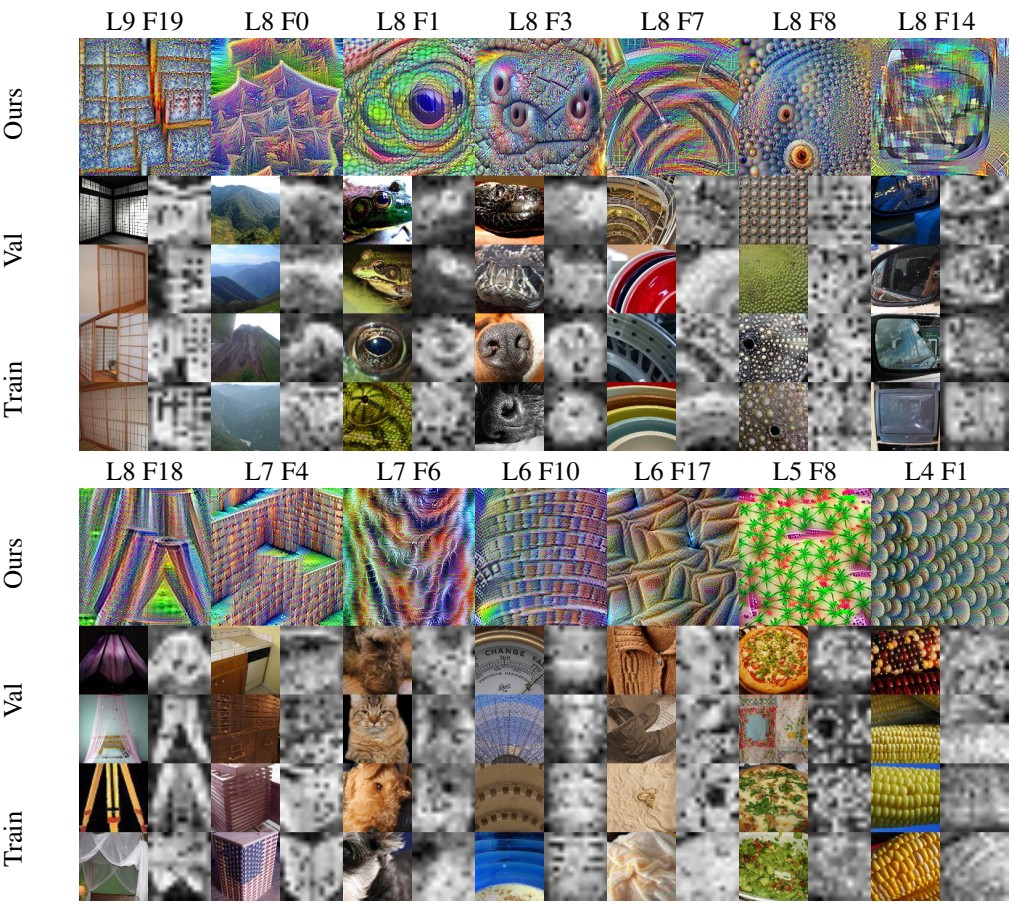

Figure 40: (Cont.) Visualization of features in ConViT small.

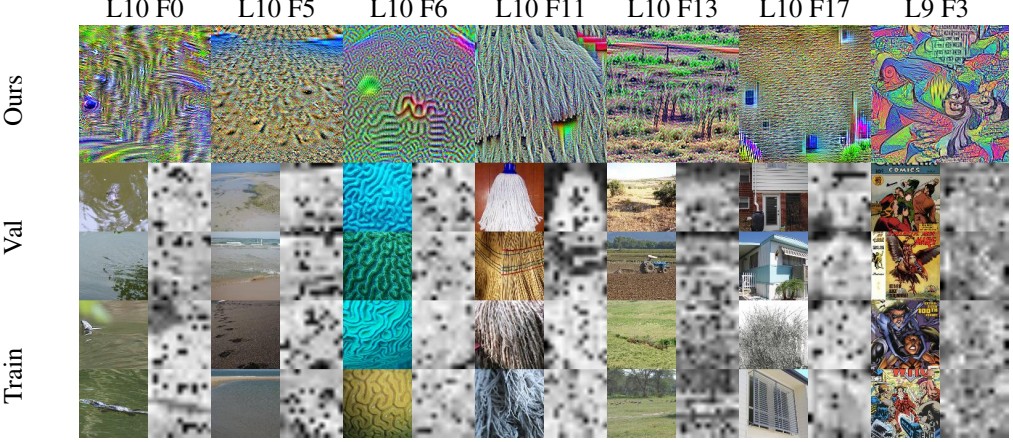

Figure 41: Visualization of features in ConViT tiny.

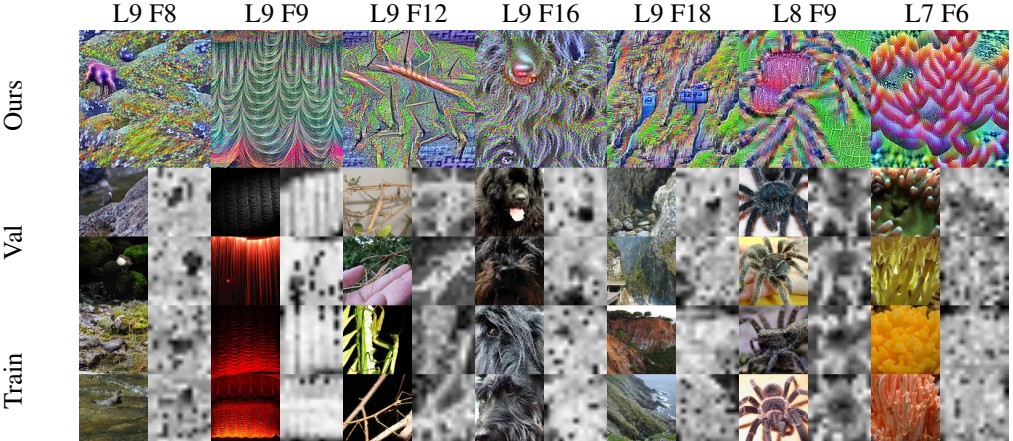

Figure 42: (Cont.) Visualization of features in ConViT tiny.

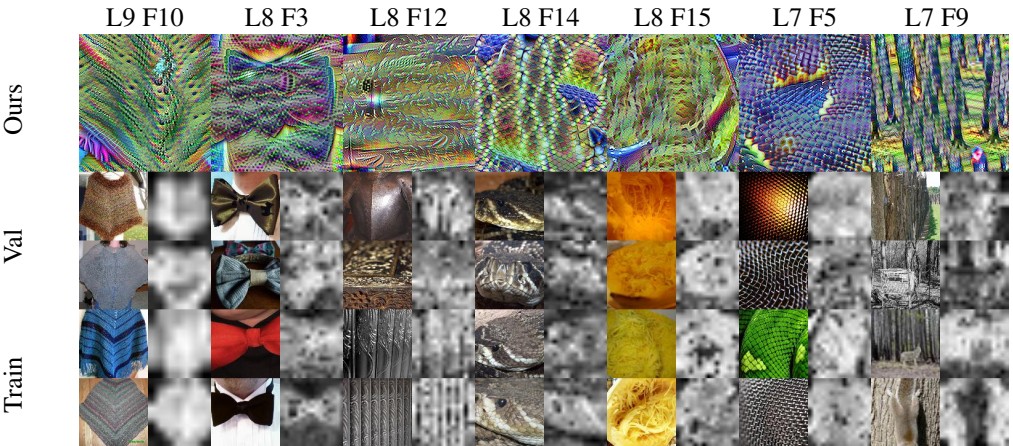

Figure 43: Visualization of features in Pit base im-224

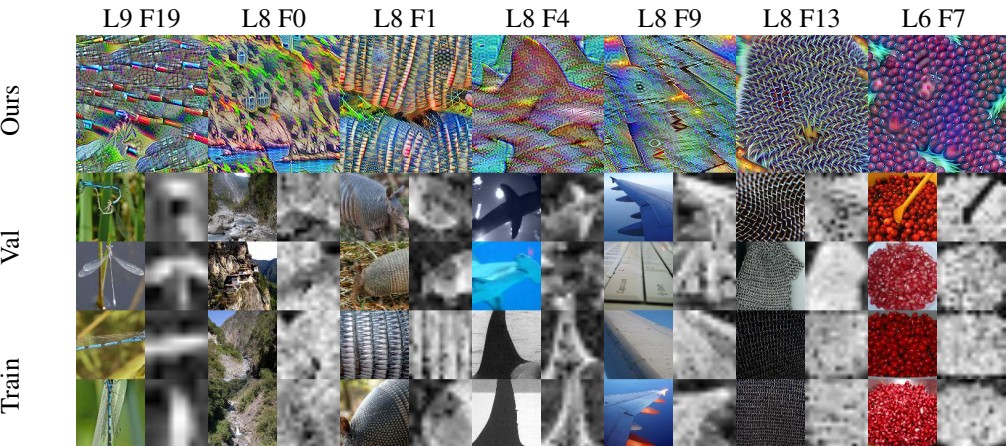

Figure 44: Visualization of features in Pit base distilled im-224.

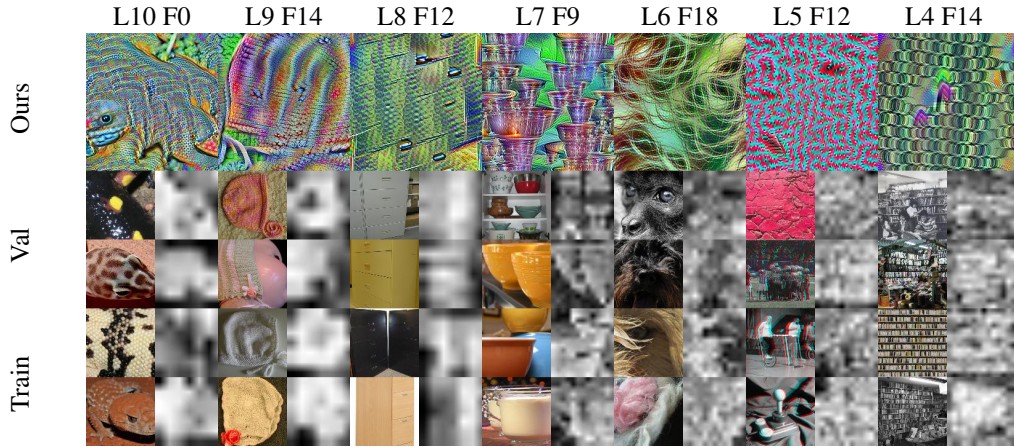

Figure 45: Visualization of features in Pit small im-224.

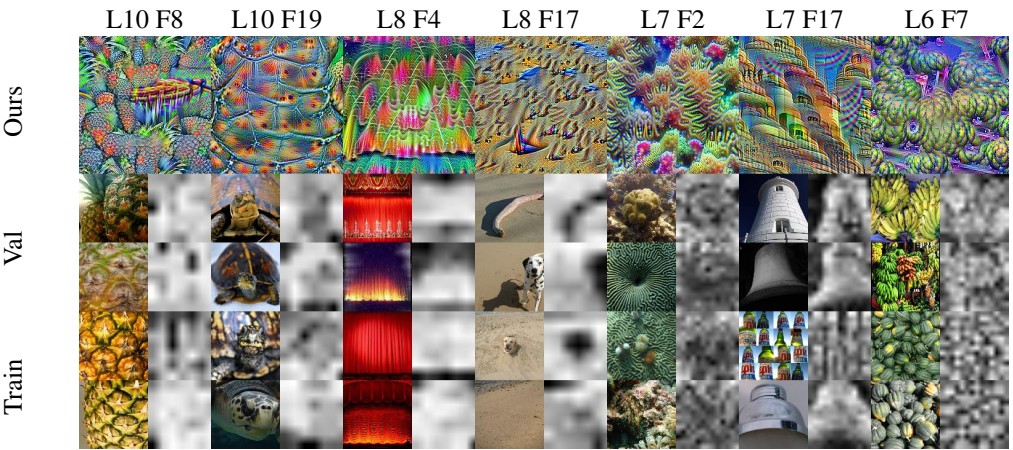

Figure 46: Visualization of features in Pit small distilled im-224.

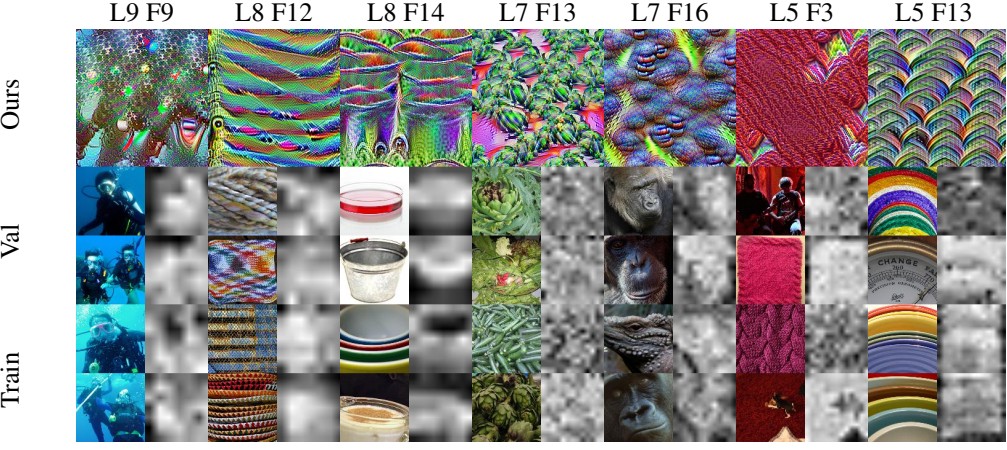

Figure 47: Visualization of features in Pit tiny im-224.

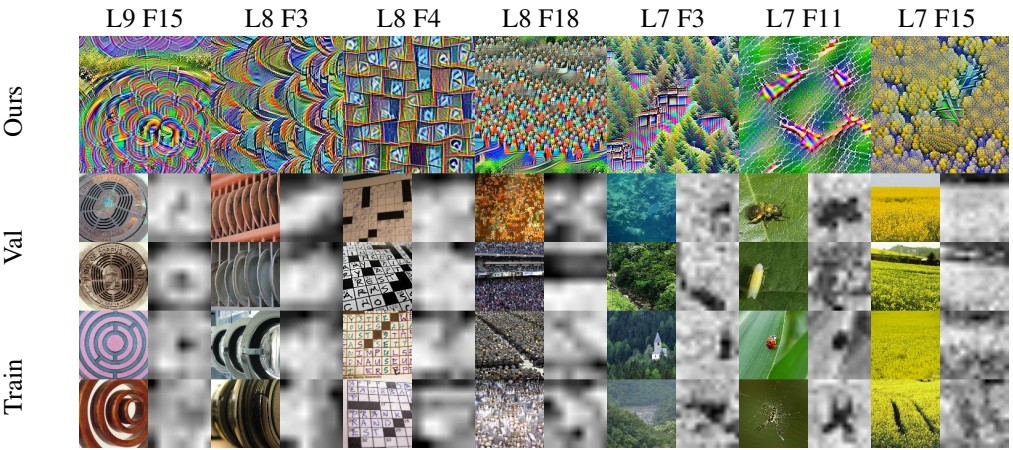

Figure 48: Visualization of features in Pit tiny distilled im-224.

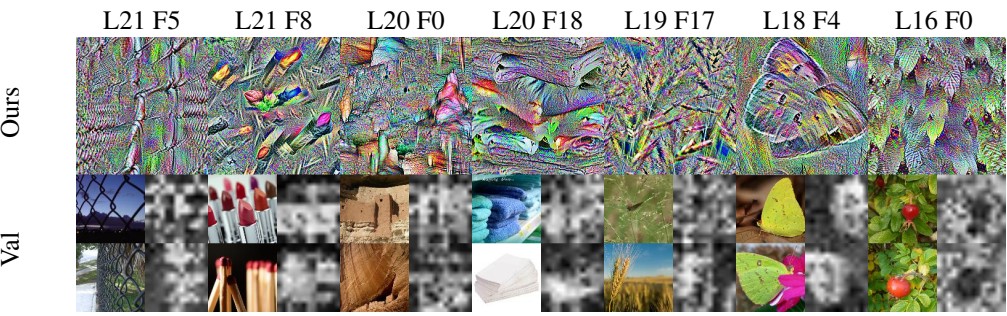

Figure 49: Visualization of features in Swin base p-4 w-7 im-224.

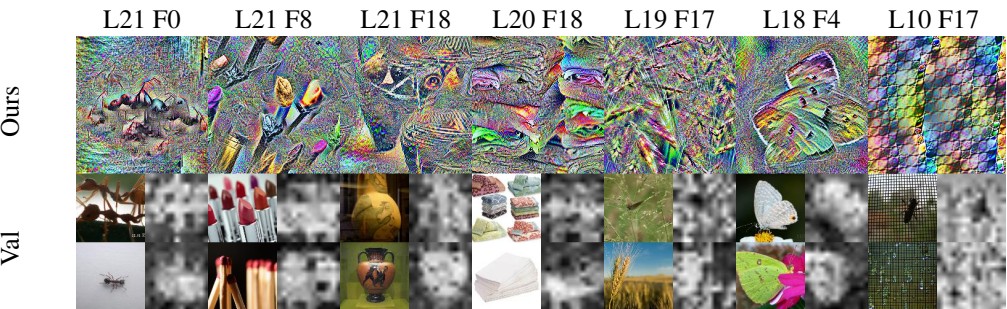

Figure 50: Visualization of features in Swing base base p-4 w-7 im-224 imagenet 22k.

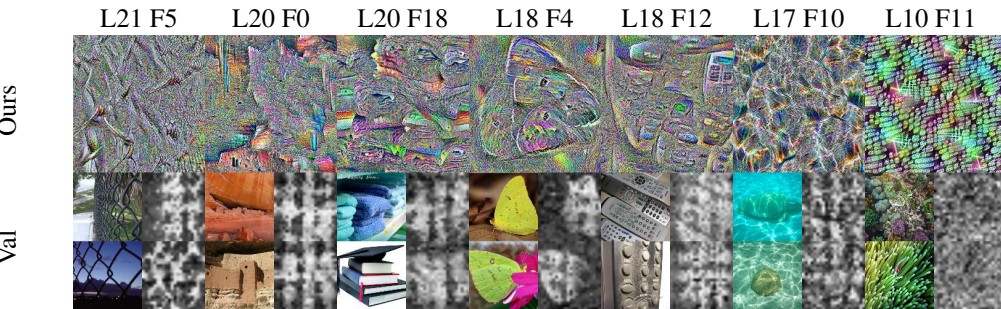

Figure 51: Visualization of features in Swin base p-4 w-12 im-384.

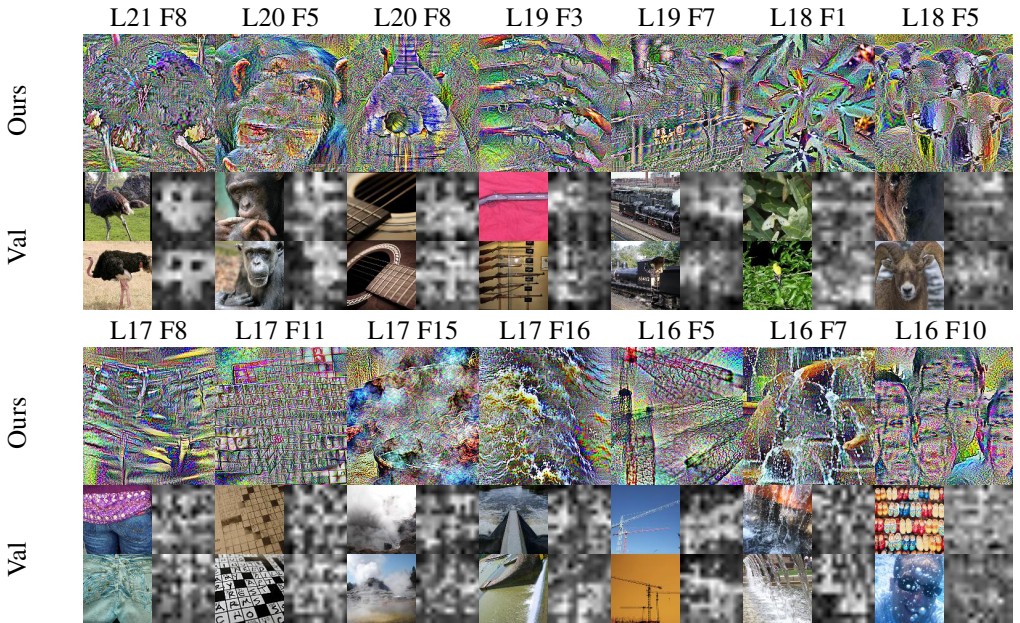

Figure 52: Visualization of features in Swin large p-4 w-7 im-224.

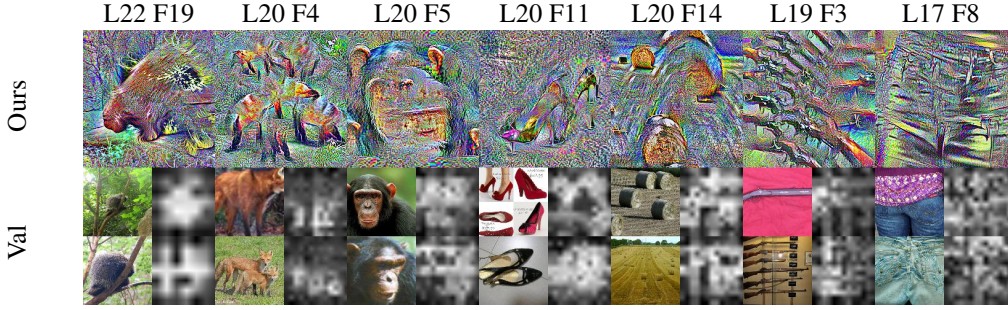

Figure 53: Visualization of features in Swin large p-4 w-7 im-224 imagenet 22k.

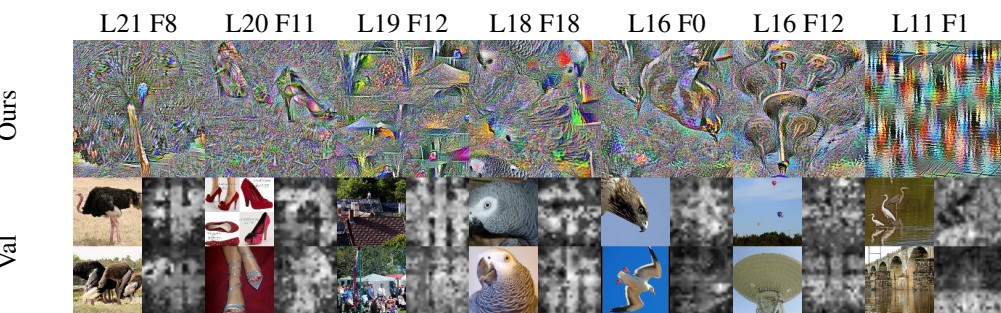

Figure 54: Visualization of features in Swin large p-4 w-12 im-384.

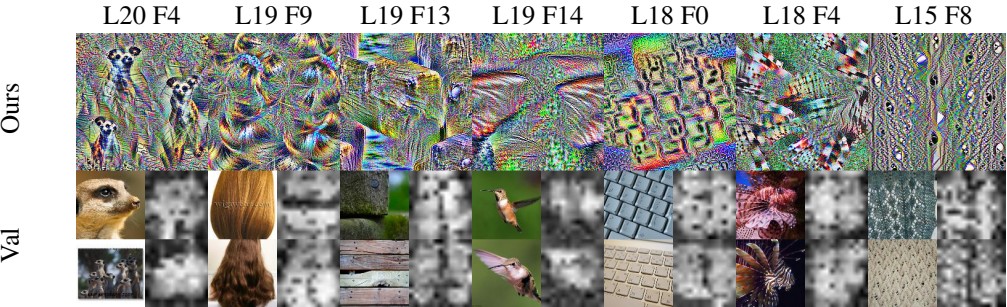

Figure 55: Visualization of features in Swin small p-4 w-7 im-224.

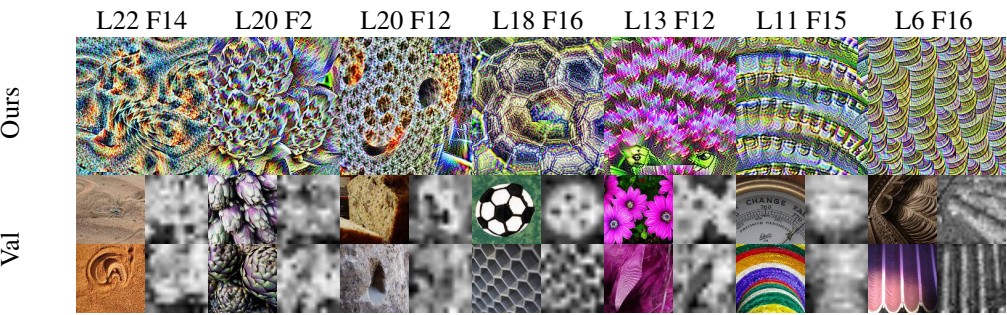

Figure 56: Visualization of features in Twins pcpvt base.

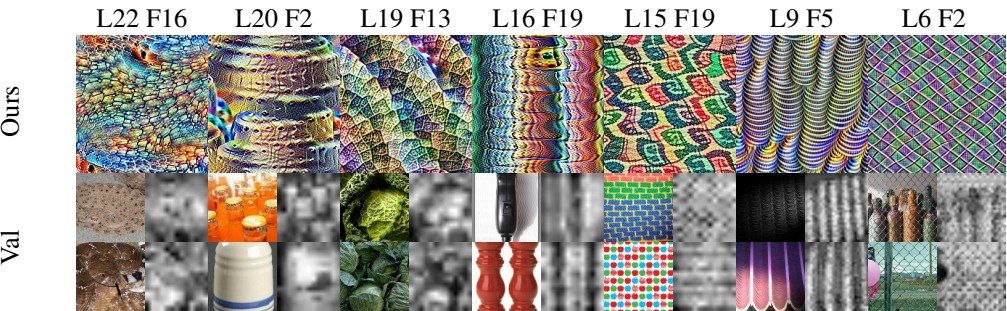

Figure 57: Visualization of features in Twins pcpvt large.

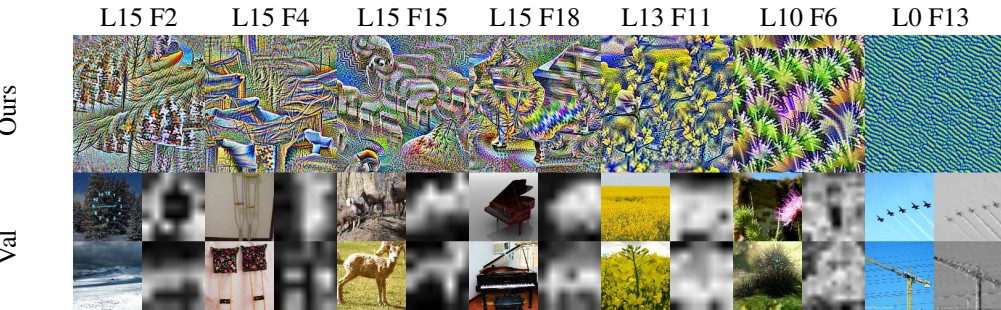

Figure 58: Visualization of features in Twins pcpvt small.

L23 F5 L21 F4 L21 F6 L21 F13 L18 F13 L17 F18 L16 F3

Ours

Val

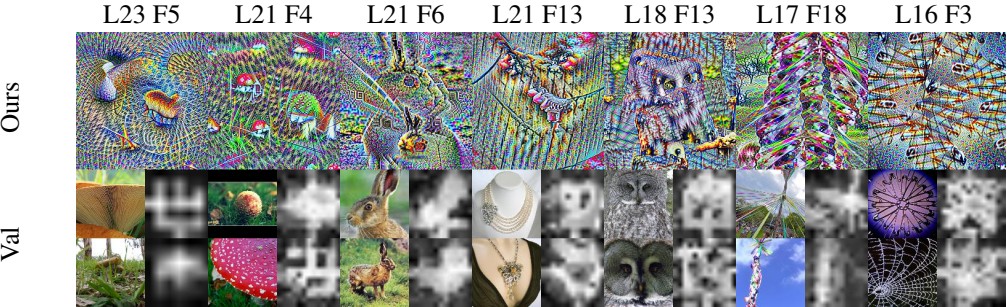

Figure 59: Visualization of features in Twins svt base.

L19 F8 L19 F13 L18 F15 L17 F5 L16 F3 L16 F11 L14 F18

Ours

Val

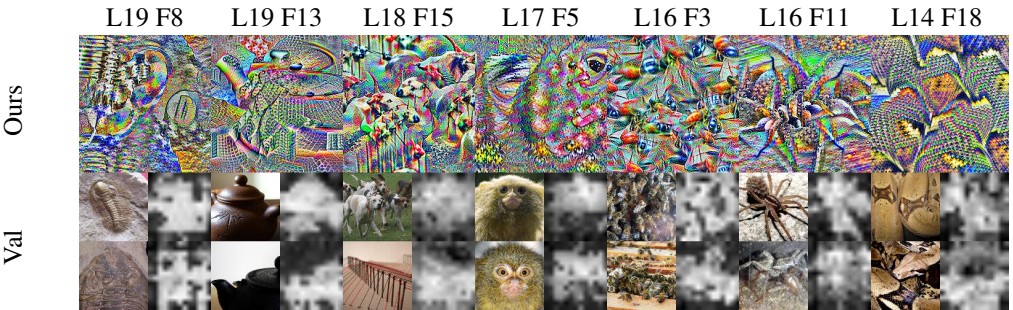

Figure 60: Visualization of features in Twins svt large.

L17 F2 L13 F5 L10 F10 L9 F3 L9 F5 L8 F7 L0 F10

Ours

Val

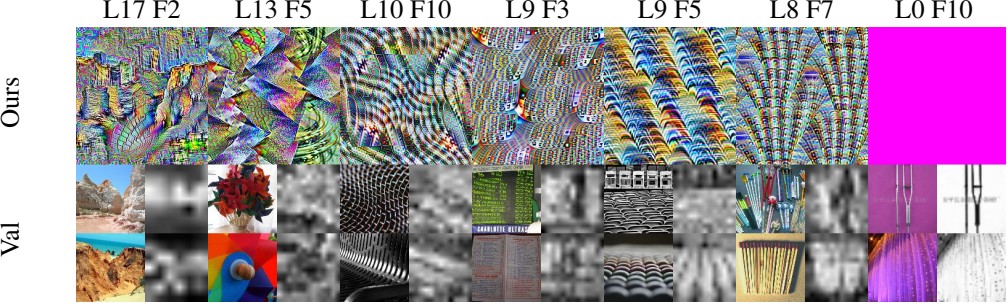

Figure 61: Visualization of features in Twins svt small.

Table 3: **ViTs more effectively correlate background information with correct class.** Both foreground and background data are normalized by full image top-1 accuracy.

| Normalized Top-1 ImageNet Accuracy | | | |
|---|---|---|---|
| Architecture | Full Image | Foreground | Background |
| ViT-B32 | 89.25 | 91.53 | 15.04 |
| ViT-L16 | 95.00 | 93.88 | **19.08** |
| ViT-L32 | 94.64 | 90.63 | 17.67 |
| ViT-B16 | 92.37 | 93.70 | 16.98 |
| ResNet-50 | 87.67 | 85.59 | 9.25 |
| ResNet-152 | 82.92 | 82.03 | 8.24 |
| MobileNetv2 | 83.77 | 85.58 | 8.75 |
| DenseNet121 | 90.58 | 86.53 | **9.72** |

## G    SPATIAL INFORMATION PRESENCE - QUANTITATIVE EVALUATION

In the following experiments, we find the most activating images for each feature. Then, we forward these images to the network. We call a patch active if the activation for this patch is higher than 0.5. First we mask out all the inactive patches meaning that we replace them with black patches. Then, we mask out x percent of the active patches in the current image. Then, we forward this new image to the network. Finally, we plot the number/sum of active patches of the modified image divided by the number/sum of the active patches in the initial image for different percentages. If the spatial information is present in a layer, we expect this number to have a linear trend. As we see in figures 62, and 63, all the layers except for the last one, have a linear trend, indicating that the loss of spatial information in individual channels is mostly made in the last layer.

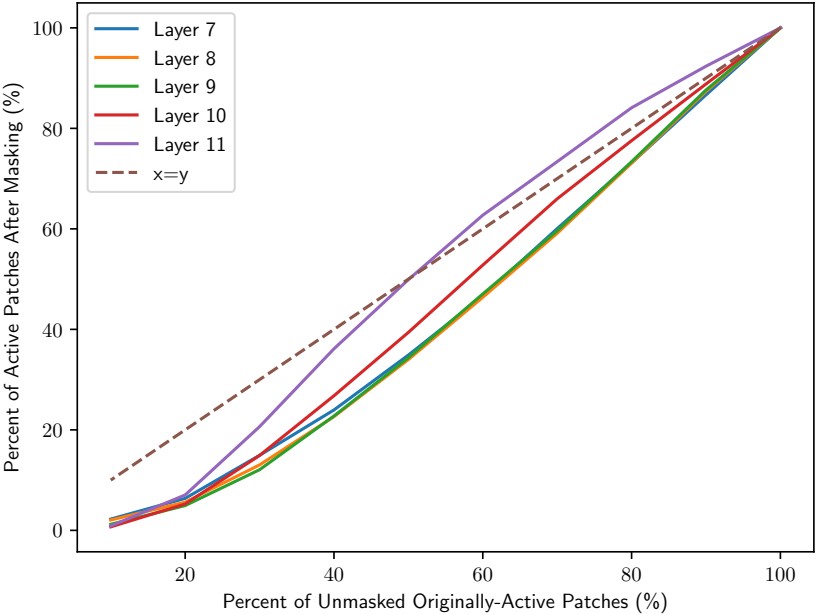

Figure 62: **Number of active patches after drop divided by number of active patches before drop**

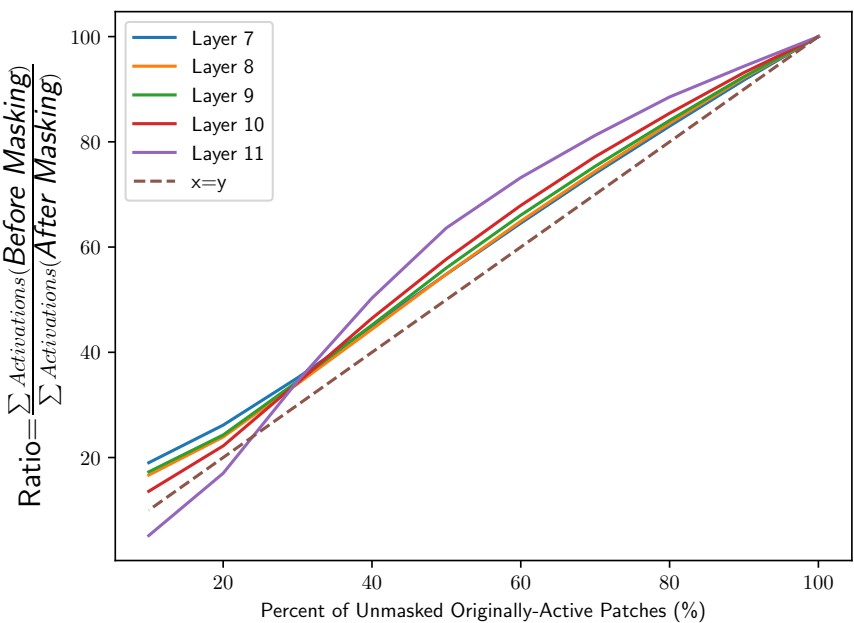

Figure 63: **Sum of active patches after drop divided by number of active patches before drop**

