# OpenReview forum: "What do vision transformers learn? A visual exploration"
_ICLR.cc/2024/Conference — Submitted to ICLR 2024_

### Official Review · Reviewer_TawR · 2023-10-24

**Soundness:** 3 good
**Presentation:** 2 fair
**Contribution:** 2 fair
**Rating:** 5
**Confidence:** 4

**Summary:**

This paper presents a framework for visualizing the features learned by vision transformers to gain insight into their internal mechanisms. It offers several key findings and conclusions, including the suitability of the linear projections in the feed-forward layers for visualization, the preservation of spatial information by ViTs, their enhanced utilization of background information compared to CNNs, and the observation that ViTs trained using CLIP learn visual concept features rather than object-specific features. The study conducts extensive experiments and visualizations to demonstrate the effectiveness of the proposed framework and to validate these findings.

**Strengths:**

- This paper is among the first few works that visualize the feature learned in vision transformers.
- A substantial number of experiments have been conducted to offer a comprehensive visual analysis.
- Some findings such as “ViTs  make better use of background information” is interesting.

**Weaknesses:**

- The visualization method employed in this study lacks novelty.
- Section 2.2 and Section 3 suffer from incomplete or missing references, and there is room for improvement in the overall writing quality.
- The results presented in Table 1 and Section 4 fail to yield significant or novel insights. A noticeable performance disparity exists between "Natural Accuracy" and "Isolating CLS." It would be expected that fine-tuning solely the linear classification head could close this performance gap, reinforcing the notion that the CLS token primarily aggregates information in later layers. It's worth mentioning that prior research has also indicated the viability of placing the CLS token in later layers (Touvron et al. 2021), and using global averages for ViTs (Liu et al. 2021). Therefore, these findings lack substantial significance.
- While the findings in Section 5 and Table 2 are interesting, they are limited to evaluating only the basic ViT architecture.
- Currently, the presented findings appear somewhat isolated. To enhance the paper's quality, it is advisable to provide a more in-depth analysis and insight into the interrelationship between these findings.


1. Touvron, Hugo, et al. "Going deeper with image transformers." ICCV 2021.
2. Liu, Ze, et al. "Swin transformer: Hierarchical vision transformer using shifted windows." ICCV 2021.

**Questions:**

- What does the term “Normalized Top-5 ImageNet Accuracy” mean?
- The choice of the fifth layer in Section 6 is not well-justified and requires further explanation.

---

> ### Author Response · Authors · 2023-11-19
> **Response**
>
> Thanks for the detailed review of our paper. We've worked on addressing all your points. If you've got more questions or thoughts, please let us know.
> >The visualization method employed in this study lacks novelty.
>
>
> Numerous techniques employed are indeed borrowed from other papers, especially “Plug-In Inversion: Model-Agnostic Inversion for Vision with Data Augmentations.” However, visualizing Vision Transformers (ViTs) has remained unexplored until now, primarily due to the challenges outlined in our paper, such as determining the optimal network position for these visualizations. Notably, OpenAI's microscope (https://microscope.openai.com/models), a tool visualizing features of various networks, lacks ViT visualizations, underscoring the intricacies associated with visualizing these particular networks.
>
> >Section 2.2 and Section 3 suffer from incomplete or missing references, and there is room for improvement in the overall writing quality.
>
> Thank you for bringing this to our attention. We have addressed and resolved the missing and incomplete references in the latest submission.
>
>
> >The results presented in Table 1 and Section 4 fail to yield significant or novel insights. A noticeable performance disparity exists between "Natural Accuracy" and "Isolating CLS." It would be expected that fine-tuning solely the linear classification head could close this performance gap, reinforcing the notion that the CLS token primarily aggregates information in later layers. It's worth mentioning that prior research has also indicated the viability of placing the CLS token in later layers (Touvron et al. 2021), and using global averages for ViTs (Liu et al. 2021). Therefore, these findings lack substantial significance.
>
>
> While adjusting the linear classification head could potentially narrow the performance gap, the key emphasis of the paper is that the aggregation of information to the CLS token is primarily conducted in the final layer, resulting in an accuracy of 78.61, as opposed to the 84.20 natural accuracy.
>
>
> Touvron et al. (2021) raise concerns about the injection of the CLS token at the beginning, emphasizing potential harm to performance. Our designed isolated CLS token serves a distinct purpose, showcasing that the last layer operates as a global average pooling layer.
>
> While using GAP, as Liu et al. (2021) suggest, is a valid alternative to the CLS token, we show that even with the CLS token, the last layer performs a similar operation to GAP.
>
> >While the findings in Section 5 and Table 2 are interesting, they are limited to evaluating only the basic ViT architecture.
>
> Thank you for your suggestion. In the latest submission, we have included other ViT models like DeiT, ConViT, and Swin in Table 2. We also added EfficientNet models, as one of the reviewers kindly suggested, for a fair comparison.
>
> >Currently, the presented findings appear somewhat isolated. To enhance the paper's quality, it is advisable to provide a more in-depth analysis and insight into the interrelationship between these findings.
> [todo]
>
> Like many interpretability and visualization papers, our paper primarily has two key elements. Firstly, we delve into how to visualize features of Vision Transformers (ViTs). Secondly, we aim to derive insights from these visualizations. For instance, through this visual analysis, we note that ViTs, similar to CNNs, possess the ability to recognize backgrounds. We additionally assess this through quantitative analysis. Noticing that ViTs surpass CNNs in performance when solely focusing on either the foreground or the background. We agree that the findings from these visualizations may be isolated, but they all have the common theme of what we can learn from the visualizations.
>
>
> >What does the term “Normalized Top-5 ImageNet Accuracy” mean?
>
> Given the differing accuracies of the models we used, we normalize them by dividing each by its respective natural accuracy; this is done to ensure a fair comparison, followed by multiplying the result by 100.
>
> >The choice of the fifth layer in Section 6 is not well-justified and requires further explanation.
>
> Thank you for bringing this to our attention. Figure 9(a) showcases a feature capturing a semantic concept, which happens to be from the fifth layer. Features like 9(b)(c) are from different layers. In the new submission, we have omitted explicit mention of the 'fifth layer' to prevent confusion.
>
>
> Please let us know if you have any more questions or concerns. If you believe we've effectively addressed your concerns, we kindly request a reconsideration of your score.

---

> > ### Comment · Reviewer_TawR · 2023-11-23
> >
> > Thank you for the thorough feedback and updated results, which partially address my concerns.
> >
> > I agree that some interesting findings are revealed in this paper, but the explanations are not very convincing to me, and some of the findings lack comprehensive analysis or explicit validation.
> >
> > After reading other reviewers' comments, I'm wondering if the current visualization method borrowed from other papers is not suitable for visualization of ViTs. Particularly, the visualization of keys, queries, and values seems ineffective in extracting interpretable features, despite their critical importance in transformer-based models.
> >
> > Overall, I think this is a borderline paper and will keep my initial rating.

---

### Official Review · Reviewer_Y5ru · 2023-10-27

**Soundness:** 3 good
**Presentation:** 3 good
**Contribution:** 3 good
**Rating:** 6
**Confidence:** 3

**Summary:**

This paper compares the learning abilities of Vision Transformer vs CNN, explores if the learning abilities changes with the language guided vision transformers such as CLIP. The authors look into optimization based Vision Transformer visualization techniques for their analysis, using gradient steps to maximize feature activations, starting from random noise. The authors experiment to identify the most activating images for each feature and then forward these images to the network. The goal is to understand the presence of spatial information in different layers of the transformer. The authors claim that most of the spatial information in individual channels is lost in the last layer. They also find that ViTs consistently outperform CNNs when information, either foreground or background, is missing

**Strengths:**

1. Finding that ViTs learn to preserve spatial information despite lacking the inductive bias of CNN
2. Finding that the ViTs spatial information is lost in the last layer
3. Authors look into text guided ViTs such as CLIP in a different way than existing work which I think is an important contribution that I see will be useful for the community in understanding future vision language models

**Weaknesses:**

1. Section 2.1 last paragraph reference missing 'Related work ?'
2. Section 3, line 4 reference missing 'augmentation ensembling ?'
3. The authors claim that ViTs learn to preserve spatial information despite lacking the inductive bias of CNN but this property disappears  from the last layer. The author seems to be not sure why (section 4, page 5). This is a key finding of the paper that needs more theory and/or experiment based proof

**Questions:**

1. Will you be able to show any experimentation and/or theoretical justification on why the last layer of ViTs loses the spatial information?

---

> ### Author Response · Authors · 2023-11-20
> **Response**
>
> Thanks for the detailed review of our paper. We've worked on addressing all your points. If you've got more questions or thoughts, please let us know.
> > Section 2.1 last paragraph reference missing 'Related work ?'
> Section 3, line 4 reference missing 'augmentation ensembling ?
>
> We appreciate your bringing this to our notice. We have addressed the issue in the latest version of our submission.
>
> > The authors claim that ViTs learn to preserve spatial information despite lacking the inductive bias of CNN but this property disappears from the last layer. The author seems to be not sure why (section 4, page 5). This is a key finding of the paper that needs more theory and/or experiment based proof
>
> That's an insightful question. We hypothesize that this characteristic arises from the network's residual connections. Spatial information is initially encoded in the first layer of the network, and these residual connections act to reinforce the retention of this information in each individual patch. We designed a quantitative experiment in Appendix G to measure the perseverance of spatial information in different layers to prove our claim.
>
>
> > Will you be able to show any experimentation and/or theoretical justification on why the last layer of ViTs loses the spatial information?
>
>
> We hypothesize that the spatial information is discarded in the last layer because the network tries to aggregate the features, and perhaps the easiest way for the network to do that is to behave like a GAP layer. In addition, we designed an experiment in Appendix G to validate the loss of spatial information in the last layer quantitatively.
>
> Please let us know if you have any more questions or concerns. If you believe we've effectively addressed your concerns, we kindly request a reconsideration of your score.

---

> > ### Comment · Reviewer_Y5ru · 2023-11-22
> >
> > Thanks to authors for providing additional explanation.
> >
> > Appendix G page 42 - you mentioned that " If the spatial information is present in a layer, we expect this number to have a linear trend".
> >
> > Why do you think the presence of spatial information is related to linearity here? It seems to me that in Figure 2, as long as 'y' increases with the increase of 'x', even if not linear increase, it is inconclusive of loss of information.
> >
> > Overall I think this paper has some interesting finding but lacks a cohesive message.
> >
> > I will keep my current rating as is.

---

### Official Review · Reviewer_ESvq · 2023-10-30

**Soundness:** 2 fair
**Presentation:** 2 fair
**Contribution:** 1 poor
**Rating:** 3
**Confidence:** 4

**Summary:**

The paper visual the intermediate features learned by ViTs using an optimization-based approach. It then explores the differences between ViTs and CNNs on the learned features and find that ViTs tend to detect background features and depend less on texture information. Further they show that ViTs maintain spatial information in all layers except the last one. The visualization is also conducted on other variants including DeiT, CoaT, Swin and Twin etc.

**Strengths:**

-	Visualization features for ViTs is an important but largely neglected topic. This work presents some solid feature visualization results and may inspire the community on related research.

**Weaknesses:**

-	The novelty of the visualization method is limited. It mainly borrows the method of Olah et al. 2017 and adapt it on ViTs with some engineering tweaks.
-	Some observations of this work are not new. For example, the authors find that ViTs maintain spatial information in all layers except the last one, and the last layer produces very similar patch tokens. This behavior has been pointed out by some existing papers. Check “DeepViT: Towards Deeper Vision Transformer 2021.” It has shown that patch tokens of the last layer are almost the same, producing very similar output patch tokens. So it is not surprising that you can just use any of the patch tokens in the last layer to predict the result and achieve decent accuracy.
-	In table2, multiple ViT models and CNN models are compared to show that the ViTs are better at using background information to predict correct classes. The issue here is that the used ViTs are more powerful than the CNNs with more parameters and more computations. ViTs have consistently better classification accuracies. The comparison is not fair and thus the conclusion of “ViTs better at using background information” is not convincing.

**Questions:**

There are some missing citations.  Sec. 2.1: Recent work ? has studied. Sec. 3: and augmentation ensembling (?).

---

> ### Author Response · Authors · 2023-11-19
> **Response**
>
> Thanks for the detailed review of our paper. We've worked on addressing all your points. If you've got more questions or thoughts, please let us know.
>
> > The novelty of the visualization method is limited. It mainly borrows the method of Olah et al. 2017 and adapt it on ViTs with some engineering tweaks.
>
>
> Numerous techniques employed are indeed borrowed from other papers, especially “Plug-In Inversion: Model-Agnostic Inversion for Vision with Data Augmentations.” However, visualizing Vision Transformers (ViTs) has remained unexplored until now, primarily due to the challenges outlined in our paper, such as determining the optimal network position for these visualizations. Notably, OpenAI's microscope (https://microscope.openai.com/models), a tool visualizing features of various networks, lacks ViT visualizations, underscoring the intricacies associated with visualizing these particular networks.
>
>
> >Some observations of this work are not new. For example, the authors find that ViTs maintain spatial information in all layers except the last one, and the last layer produces very similar patch tokens. This behavior has been pointed out by some existing papers. Check “DeepViT: Towards Deeper Vision Transformer 2021.” It has shown that patch tokens of the last layer are almost the same, producing very similar output patch tokens. So it is not surprising that you can just use any of the patch tokens in the last layer to predict the result and achieve decent accuracy.
>
>
> There are several key distinctions between the findings presented in our paper and those outlined in DeepViT. To the best of our knowledge, DeepViT does not demonstrate uniformity in the patch embeddings of the last layer. Their emphasis lies in illustrating that attention maps tend to become more uniform as one progresses deeper into the network. Examining the first row of Figure 6 in DeepViT, specifically for a ViT model trained without their re-attention technique, a diagonal line is observable, indicating that the attention map is not entirely uniform. This suggests a reduced effectiveness of the attention map in the final layers, causing the model to resemble an MLP model. However, DeepViT does not establish that the patch features are identical in the last layer.
> Moreover, DeepViT compares features between the final layer and preceding layers, asserting a diminishing evolution of features. They argue that the overall set of features becomes more alike in the ultimate layers. For example, the features of the last layer are alike the features of the layer before that. However, it's essential to clarify that this observation does not automatically suggest identity among the patches in the final layer.
>
> > In table2, multiple ViT models and CNN models are compared to show that the ViTs are better at using background information to predict correct classes. The issue here is that the used ViTs are more powerful than the CNNs with more parameters and more computations. ViTs have consistently better classification accuracies. The comparison is not fair and thus the conclusion of “ViTs better at using background information” is not convincing.
>
>
> Thank you for your helpful suggestion. In the latest submission, we added three EfficientNet models with accuracies on par or even better than some of the other ViT models we used. We also added DeiT, ConViT, and Swin to this table, as one of the reviewers kindly suggested. Also, the accuracies presented in Table 2 are normalized. This means we divided the accuracy by the model's accuracy and multiplied the result by 100.
>
>
>
> >There are some missing citations. Sec. 2.1: Recent work ? has studied. Sec. 3: and augmentation ensembling (?).
>
>
> Thank you for bringing this to our attention. We have addressed and resolved the issue in the latest submission.
>
>
> Please let us know if you have any more questions or concerns. If you believe we've effectively addressed your concerns, we kindly request a reconsideration of your score.

---

### Official Review · Reviewer_f2Nw · 2023-11-01

**Soundness:** 3 good
**Presentation:** 3 good
**Contribution:** 2 fair
**Rating:** 5
**Confidence:** 4

**Summary:**

In this work, authors  address  the visualizations on vision transformers. they observe build up of complex features across the hierarchy across layers, similar to CNNs. They also find that neurons in ViTs trained with language model supervision (e.g., CLIP) are activated by semantic concepts rather than visual features. They also show that transformers detect image background features, just like CNNs.

**Strengths:**

This work addresses an important problem in deep learning which is understanding how visual transformers work, and shed light on these black boxes. This is certainly helpful to, in particular, the vision community.

The paper is generally well organized and well written. The prior research is also adequately mentioned.

The findings, although, not all being quite novel, are interesting. In particular, I find the finding that transformers make better use of background and foreground information, compared to CNNs, interesting. "we find that transformers detect image background features, just like their convolutional counterparts, but their predictions depend far less on high-frequency information."


This works applies classic methods for visualizing CNNs to ViTs and is that sense is actually not very different from other approaches and results are somewhat expected and less surprising. The works emphasizes on similarities between CNNs and Transformers, in particular progressive specialization, rather than highlighting main differences deep inside the network, rather what information is being used.

**Weaknesses:**

The work still does not get into the meat of what transformers really do! For example, what key, query, and value do? and what makes them more effective. For example, it is shown that they use foreground and background more effectively, but it is not explored why that happens. Another important aspect would be how the key,query,value operations relate to convolution. Some visualization may help get insights regarding this.


Minor issues:
Page 3  Recent work ? —> missing reference

**Questions:**

Q: in light of your results can you tell what really explains higher performance of vision transformers compared to CNNs? I mean in the architecture? can that be visualized?

Q: how much of these also apply to the MLPmixer architecture, or its variants

Q: It seems like authors, are discarding visualization of keys, queries and values! I think visualization of these features might actually gives good insights regarding the designs of ViT. And to understand how exactly these differ from convolution operation!

Q: In page 5, section: you are discussing the receptive field of neurons. It is true that neurons in all layers have full receptive field in the image, but what is the effective receptive field? In other words, can you show how much and where in the image a neuron in getting its input from?

Q: For long researchers believed that CNNs might be a good model of how vision should be done mainly because they parallel well with human visual system and extract features in a hierarchy. Vision transformers seem to match less with CNNs and human visual system in terms of their structure. Can we say we are diverging from coming up with a unified model of how vision should be done?

---

> ### Author Response · Authors · 2023-11-19
> **Response**
>
> Thanks for the detailed review of our paper. We've worked on addressing all your points. If you've got more questions or thoughts, please let us know.
>
> > Minor issues: Page 3 Recent work? —> missing reference
>
> Thank you for bringing this to our attention. We have addressed and resolved the issue in the latest submission.
>
> > How much of these also apply to the MLPmixer architecture or its variants
>
> We think this falls outside our project's scope, as our focus was on investigating Vision Transformers (ViTs). However, if you think that exploring MLPs also might be a good scope for this paper, please let us know, and we can perform these visualizations on MLPs.
>
>
> > It seems like authors, are discarding visualization of keys, queries and values! I think visualization of these features might actually gives good insights regarding the designs of ViT. And to understand how exactly these differ from convolution operation!
>
> In the final paragraph of page 4, we explore the visualization of keys, queries, and values. Unfortunately, this visualization fails to extract interpretable features and to distinguish between early and deep layers. Despite visualizing many keys, queries, and values, the results do not yield clear and interpretable visualizations. Figure 3 provides an example of such visualizations for reference.
>
> > In page 5, section: you are discussing the receptive field of neurons. It is true that neurons in all layers have full receptive field in the image, but what is the effective receptive field? In other words, can you show how much and where in the image a neuron in getting its input from?
>
> A quantitative analysis of the effective receptive field can be found in the paper titled "Do Vision Transformers See Like Convolutional Neural Networks?" The visualization of the effective receptive field is presented in Figure 6 within the same paper, revealing that the effective receptive field expands with increasing network depth. The discussion in the initial paragraph of section 4 explores the preservation of spatial information within individual channels of Vision Transformers (ViTs) despite the potential for ViTs to have a full receptive field even in their initial layers. We hypothesized that this preservation stems from the residual connections inherent in ViTs.
>
> >Q: For long researchers believed that CNNs might be a good model of how vision should be done mainly because they parallel well with human visual system and extract features in a hierarchy. Vision transformers seem to match less with CNNs and human visual system in terms of their structure. Can we say we are diverging from coming up with a unified model of how vision should be done?
>
> Vision Transformers (ViTs) might emerge as the preferred models for vision-related tasks despite their absence of inductive bias and deviation from the functioning of the human visual system. Nonetheless, as we discuss in the paper, they do encompass certain biases, such as transitioning from local to global features. Additionally, their versatility in handling Natural Language Processing (NLP) tasks makes them a favorable option for multi-modal tasks. Hence, we believe they could become the go-to models for Vision Tasks.
>
> Please let us know if you have any more questions or concerns. If you believe we've effectively addressed your concerns, we kindly request a reconsideration of your score.

---

### Author Response · Authors · 2023-11-21
**To Dear Reviewers**

Dear Reviewers,

As the discussion period ends, we would like to kindly ask for your reply to our remarks. We feel we have addressed your concerns and we look forward to further discussions and guidance. Please let us know if there is anything else we can do or other questions we can answer.

---

### Meta-Review · Area_Chair_x4Bb · 2023-12-04

**Metareview:**

The paper visualize the intermediate features learned by ViTs. It then explores the differences between ViTs and CNNs on the learned features and find that ViTs tend to detect background features and depend less on texture information. Further they show that ViTs maintain spatial information in all layers except the last one.

**Justification For Why Not Higher Score:**

- Most of the reviewers agree that this work only directly borrows the feature visualization technique from CNNs, and apply it for ViT feature visualization. It does not show key features learned by some unique modules of ViT models, like the key, value and query. This makes the novelty of this paper limited.

- Some observations of this work are not new. For example, the authors find that ViTs maintain spatial information in all layers except the last one, and the last layer produces very similar patch tokens. This behavior has been pointed out by some existing papers, like “DeepViT: Towards Deeper Vision Transformer 2021.”

**Justification For Why Not Lower Score:**

NA

---

### Decision · Program_Chairs · 2024-01-16

Reject